# High-Quality Perovskite Thin Films for NO_2_ Detection: Optimizing Pulsed Laser Deposition of Pure and Sr-Doped LaMO_3_ (M = Co, Fe)

**DOI:** 10.3390/ma18051175

**Published:** 2025-03-06

**Authors:** Lukasz Cieniek, Agnieszka Kopia, Kazimierz Kowalski, Tomasz Moskalewicz

**Affiliations:** Faculty of Metals Engineering and Industrial Computer Science, AGH University of Krakow, al. Mickiewicza 30, 30-059 Krakow, Poland

**Keywords:** perovskite, thin films, pulsed laser deposition (PLD), gas sensor, NO_2_, LaCoO_3_, LaFeO_3_, Sr doping

## Abstract

This study investigates the structural and catalytic properties of pure and Sr-doped LaCoO_3_ and LaFeO_3_ thin films for potential use as resistive gas sensors. Thin films were deposited via pulsed laser deposition (PLD) and characterized using X-ray diffraction (XRD), X-ray photoelectron spectroscopy (XPS), scanning electron microscopy (SEM), atomic force microscopy (AFM), nanoindentation, and scratch tests. XRD analysis confirmed the formation of the desired perovskite phases without secondary phases. XPS revealed the presence of La^3+^, Co^3+^/Co^4+^, Fe^3+^/Fe^4+^, and Sr^2+^ oxidation states. SEM and AFM imaging showed compact, nanostructured surfaces with varying morphologies (shape and size of surface irregularities) depending on the composition. Sr doping led to surface refinement and increased nanohardness and adhesion. Transmission electron microscopy (TEM) analysis confirmed the columnar growth of nanocrystalline films. Sr-doped LaCoO_3_ demonstrated enhanced sensitivity and stability in the presence of NO_2_ gas compared to pure LaCoO_3_, as evidenced by electrical resistivity measurements within 230 ÷ 440 °C. At the same time, it was found that Sr doping stabilizes the catalytic activity of LaFeO_3_ (in the range of 300 ÷ 350 °C), although its behavior in the presence of NO_2_ differs from that of LaCo(Sr)O_3_—especially in terms of response and recovery times. These findings highlight the potential of Sr-doped LaCoO_3_ and LaFeO_3_ thin films for NO_2_ sensing applications.

## 1. Introduction

With the continued expansion of industrial activities and the associated increase in anthropogenic impact on the natural environment, air pollution control has become a critical challenge today. Increased public awareness brought the issue of air pollution to the forefront of public discourse. Gas detection instruments have become indispensable tools in many industrial sectors. Modern gas sensors facilitate the detection of a range of gaseous pollutants, including but not limited to CO, NO, NO_2_, NH_3_, SO_2_, and CO_2_ [1]. These gases have detrimental effects on human health [2], underlining the importance of public access to air quality information. Gas sensors also found widespread applications in industries such as automotive (lambda probes, SCR systems [3]), aviation [4], energy [5], and medicine (disease diagnostics) [6].

To reduce the harmful environmental impact of vehicle emissions, the European Union introduced strict regulations on the permissible levels of harmful gases released during combustion. The introduction of increasingly stringent Euro standards forced automakers to develop more efficient engines that minimize pollutant emissions. In diesel engines, a key technology used to achieve this goal is the selective catalytic reduction (SCR) system [7]. A critical component of the SCR system is the NO_2_ sensor, typically located in the exhaust stream. This sensor plays a vital role in accurately measuring the concentration of nitrogen dioxide (NO_2_) in the exhaust gases. The sensor data are then transmitted to the engine control unit (ECU), which uses this information to control the injection of urea solution (AdBlue) into the exhaust stream. AdBlue initiates a chemical reaction that converts harmful NO_2_ into benign nitrogen (N_2_) and oxygen (O_2_) [8]. The effectiveness of the SCR system depends on the fast, reliable, and accurate determination of the exhaust gas composition by the NO_2_ sensor. Accurate measurement of NO_2_ concentration is essential to ensure the optimal amount of AdBlue is injected to maximize NO_2_ reduction while preventing ammonia slip and urea crystallization [9]. Lambda probes measure the oxygen content of fuel–air mixtures to optimize combustion and reduce emissions. The energy industry also needs gas sensors, especially in power plants where high temperatures and pressures require durable and reliable sensors [10]. Traditional thick film sensors have limitations in sensitivity and response time [11]. Thin film technology offers improvements but can suffer from stability issues. Promising perovskite materials offer stability and good electrical properties [12].

This work focuses on the analysis of the structure and properties of pure and Sr-doped LaCoO_3_ and LaFeO_3_ thin films for use in resistive sensors. Perovskites received considerable attention from researchers due to their ability to influence the physical properties of a material through electron–electron and electron–phonon interactions. These materials exhibit a wide range of phenomena, including high-temperature superconductivity, bandgap ferromagnetism, and multiple physical properties within a single crystal. Perovskites containing Mn [13], Co [14], or Ru [15] are particularly noteworthy in this regard.

## 2. Research Material

A gas sensor is a device that detects and quantifies the presence or concentration of a gas; it translates chemical information about the gas into a measurable signal. Gas sensors are commonly used to detect harmful gases such as carbon monoxide (CO), nitrogen oxides (NO_x_), hydrogen (H_2_), ammonia (NH_3_), sulfur dioxide (SO_2_), and hydrogen sulfide (H_2_S). The type of sensor used determines its operating mechanism. Gas sensors can be divided into several categories: electrochemical gas sensors (EGS), resistance sensors (RS), optical sensors, surface acoustic wave sensors, and calorimetric sensors [16]. Regardless of the operating mechanism, ideal gas sensors should have the following characteristics: high sensitivity, selectivity to the target gas, fast response time in the presence of the target gas, fast regeneration time, ability to operate in a wide range of temperatures, and stability. The general operating principle of a gas sensor involves the interaction between a sensing electrode (SE) and the target gas. This interaction generates an electrical signal that is then processed and converted into a usable signal for analysis. A typical gas sensor consists of several key components: a gas-sensing electrode (SE), a support material (e.g., Si, MgO, and YSZ epi-polished single crystals), a heating system, and electrodes [17].

Due to the high material property demands on sensing electrodes (SE), scientists are paying special attention to perovskites. This family of materials shares a crystallographic structure akin to the mineral CaTiO_3_. While the ideal perovskite structure is cubic, with a general formula of ABX_3_, this form is not often found in nature. Most perovskites exhibit some degree of deformation. These materials attracted significant scientific interest for many years, with initial research conducted by Goldschmidt et al. in 1920 [18]. The deformations result in reduced symmetry, which profoundly impacts the magnetic and electrical properties of perovskites. Their mixed ionic and electronic conductivity led to diverse applications, including electrical industry [19], gas sensors [20], automotive industry [21], and solid oxide fuel cells (SOFCs) [22].

The perovskite cell is shown in Figure 1. The A cation in the center of the cube is surrounded by six oxygen ions (Figure 1a), forming a BO6 octahedron (Figure 1b). Smaller ions occupy the corners of the cell. While Figure 1 illustrates the structure of an ideal perovskite, real-world instances of this cell often exhibit distortions stemming from disparities in ionic radius. These mismatches induce the displacement of ions from their optimal positions. Element A is typically a large metal cation, often an alkaline earth or lanthanide, such as La, Li, Be, or Ca. Element B is typically a smaller cation, such as Ti, Nb, Ta, Mn, Fe, or Co. B ions are coordinated by six X ions, which are typically oxygen or fluorine. Perovskites containing transition metal ions as element B exhibit interesting electrical and magnetic properties.

Thermal activation of LaCoO_3_ oxides leads to changes in their magnetic and electrical properties, which is related to the spin moment of the Co^3+^ ions. As the temperature increases, the spin moment of the Co ions changes from low (LS, up to approx. 100 K), to intermediate (IS, above 100 K), to high (HS, from approx. 500 K) [23]. At low temperatures, the oxide exhibits the characteristics of a non-magnetic insulator, while at high temperatures, it exhibits the characteristics of a metallic paramagnet. Furthermore, these materials undergo crystal structure transformation also due to heating (ferroelastic phase transition—Figure 1c). At room temperature, these materials crystallize in an orthorhombic structure. Upon heating, they undergo a structural transformation to a regular structure at temperatures between 1100 °C and 1600 °C for LaCoO_3_ [24]. It has also been observed that in non-stoichiometric perovskites, the addition of Sr or Ca lowers the transformation temperature. This structural change is accompanied by the formation of transformation twins (Figure 1d), which relieve stress and explain the observed ferroelastic properties of these materials. The unique properties of these materials stem from the partially filled 3d electron shells of the transition metal ions. In the present study, two base perovskites LaMO_3_ (M = Co, Fe) and their Sr-doped variants were used as sensing electrode (SE) materials.

Lanthanum ferrite (LaFeO_3_), a perovskite material, exhibits excellent antiferromagnetic, ferromagnetic, ferroelastic, and catalytic properties, along with high ionic conductivity. It is utilized in various forms, including powders [25] and thin films [26], for sensing elements (SE) in gas sensors. Research by Dai et al. [27] demonstrated that Fe-O bonds are more active toward gas molecules due to faster oxygen reduction on the Fe-O surface with (010) orientation compared to La-O bonds in LaFeO_3_ films. LaFeO_3_ boasts the highest Neel temperature among the entire orthoferrite family. As Lyubutin et al. [28] demonstrate, the Fe-O-Fe bonds not only induce distortion of the unit cell, but also contribute to the high Neel temperature. This indicates that LaFeO_3_ is a promising material for high-temperature applications. Lanthanum ferrite (LaFeO_3_) is frequently doped with divalent elements, such as Sr, Co, Ca, Ba, Cu, Li, Mn, and Zn, substituting for the La^3+^ ions. This substitution alters the oxidation state of iron, inducing a change from Fe^3+^ to Fe^4+^ to maintain charge neutrality. Consequently, oxygen vacancies are generated to compensate for the charge imbalance, significantly impacting the electrical conductivity of the material.

Lanthanum cobaltite (LaCoO_3_) stands out as a compound with exceptional electrical, catalytic, and magnetic properties. Its remarkable mixed conductivity, characterized by high ionic and electrical conductivity, makes it suitable for applications such as cathode material in solid oxide fuel cells (SOFCs), oxygen membranes, and CO catalysts [29]. LaCoO_3_ crystallizes in either a rhombohedral (R3c) or orthorhombic (Pbnm) structure (Figure 1), a form it retains up to its melting point of 1740 °C. The rhombohedral distortion in LaCoO_3_ decreases with increasing temperature and cation concentration at the B site. The presence of mixed Co^3+^ and Co^4+^ valence states influences the electrical properties. Furthermore, the spin state of the Co ions can be modified not only by temperature, but also by changes in gas pressure that alter the magnetic properties of the material by changing interatomic distances, bond lengths, and angles [24]. This ability to alter the spin state is unusual and not observed in other magnetic oxide materials [30]. In the range of 0 to 100 K, Co^3^⁺ ions exhibit a low-spin state with a resultant spin of 0. Above 100 K, Co^3^⁺ ions gradually transition from a low-spin to an intermediate-spin state. This transition requires energy to overcome the band gap, an effect unique to this group of materials. LaCoO_3_ is an n-type semiconductor. Electrical conductivity arises from charge exchange between Co-O-Co bonds, increasing ionic conductivity by three orders of magnitude [31]. Within the temperature range of 110 K to 350 K, electron excitation from a narrow valence band to localized states at high-spin cobalt sites generates mobile small-polaron holes and leads to electron trapping at stationary Co^2^⁺ ions [32]. Upon exceeding 650 K, a metallic phase stabilizes, characterized by high-spin Co^3^⁺ ions and intermediate-spin Co(III) with a partially filled electron shell that facilitates p-type conductivity. Oxygen ion diffusion in LaCoO_3_ is attributed to the presence of vacancies [33]. LaCoO_3_ exhibits potential as a material for low-temperature CO detection [34].

The preceding discussion of the characteristics of perovskites in the context of their use as gas-sensitive materials fully justifies the decision to produce and thoroughly analyze the structure, selected mechanical properties, and catalytic properties of La(Sr)CoO_3_ and La(Sr)FeO_3_ thin films. For thin functional films, selecting an appropriate manufacturing technique is critical. Laser ablation offers a compelling solution due to its ability to transfer the phase and chemical composition from a target to a nanocrystalline thin film without gas phase decomposition. This requires considerable experience in selecting the appropriate substrate and, more importantly, optimizing several key PLD process parameters: the energy of individual laser pulses, which dictates the power density at the target surface (factoring in optical path losses), the target–substrate distance, the working gas pressure within the chamber, and the substrate temperature. This paper discusses these aspects clearly, providing a guide for depositing high-quality perovskite thin films using the PLD technique.

## 3. Methodology and Research Techniques

### 3.1. Target Fabrication for PLD Process

The initial research phase focused on meticulously preparing targets from specialized materials for subsequent pulsed laser deposition (PLD). This involved procuring commercially available materials (e.g., LaFeO_3_) and employing mechanical synthesis to create appropriate mixtures of base powders (La_2_O_3_, Co_3_O_4_, Fe_3_O_4_, and SrO). To prepare the target disks, high-energy grinding was conducted in a planetary ball mill (Retsch PM 100). This process lasted 28 ÷ 30 h at a rotational speed of 550 ÷ 650 RPM, utilizing 2 h intervals with 30 min rest periods. The synthesis employed pre-roasted base powders to eliminate moisture and ensure the formation of the desired perovskite phase with specific stoichiometry. This procedure yielded the following perovskites: stoichiometric LaCoO_3_, two Sr-doped variants (La_0_._8_Sr_0_._2_CoO_3_, La_0_._9_Sr_0_._1_CoO_3_), and two non-stoichiometric Sr-doped lanthanum ferrite powders (La_0_._8_Sr_0_._2_FeO_3_, La_0_._9_Sr_0_._1_FeO_3_). The resulting nanopowders were cold-pressed in a 1-inch diameter die with approximately 3 g (~5 droplets) of polyvinyl alcohol (at 5 MPa pressure) and sintered in an Ar atmosphere between 200 and 1200 °C. Slow cooling to ambient temperature prevented shrinkage. This procedure resulted in high-quality targets (Figure 2) with densities and porosities comparable to commercially available disks. Material quality was assessed using scanning electron microscopy (SEM) and energy-dispersive X-ray spectroscopy (EDS) to confirm chemical composition.

### 3.2. Perovskite Thin Film Deposition

Perovskite thin films, LaMO_3_ (M = Co, Fe) and their Sr-doped variants, were deposited via pulsed laser deposition (PLD) on epi-polished single-crystal Si and MgO substrates with [001] orientation. The Neocera PLD/PED system equipped with a high-energy Nd:YAG laser (Powerlite Precision II DSL 9010) and a Pioneer 180 vacuum chamber was employed for thin film fabrication (Figure 3). Table 1 summarizes details and relevant deposition parameters.

The parameters employed yielded high-quality thin films suitable for further research and analysis. The deposition process can be further optimized by slightly reducing the energy per pulse using Q-switched delay control (to about 65 ÷ 75 mJ) and simultaneously increasing the evaporation time—effectively increasing the number of pulses to about 150,000. Alternatively, increasing the target–substrate distance may be beneficial. However, it is crucial to consider that pulsed laser deposition is a directional process; an excessive working distance will significantly impact the film’s homogeneity.

### 3.3. Research and Analytical Techniques Applied

Microstructural characterization and chemical composition analysis of the thin films were conducted using scanning electron microscopy (FEI Nova NanoSEM 450, equipped with an EDAX Energy Dispersive Spectroscopy Detector SDD-APOLLO X and Octane Elect Plus by Thermo Fisher Scientific Inc., Waltham, MA, USA) and transmission electron microscopy (Jeol 200CX and JEM-2010ARP both by JEOL Ltd., Tokyo, Japan with EDX INCA Oxford Instruments by Abingdon, Oxfordshire, UK). TEM investigations were conducted on cross-sectional thin films prepared by dimpling and ion beam milling using a Gatan precision ion polishing system (PIPS). Additional lamellae were prepared by focused ion beam (FIB) milling.

The crystalline structure of the thin films was examined by X-ray diffraction (PANanalytical EMPYREAN DY 1061 by Malvern Panalytical Inc., Westborough, MA, USA) with Cu_Kα_ radiation (λ_Cu_ = 0.154 nm) at a grazing incidence angle α = 1°. The results were interpreted using PANalytical Highscore 4.9 software. Phase analysis and crystallite size determination were performed using the MAUD (Materials Analysis Using Diffraction) program version 2.999. Phase identification was achieved using the PDF-4+ database from ICDD.

Surface topography, roughness parameters, mechanical properties, and adhesion forces of the deposited perovskite thin films were evaluated using atomic force microscopy (Bruker Dimension Icon SPM System with NanoScope V Controller by Bruker Corporation, Billerica, MA, USA) as well as scratch and nanoindentation tests (CSM Instruments NHT-NST by CSM Instruments SA, Peseux, Switzerland/Anton Paar GmbH, Graz Austria).

X-ray photoelectron spectroscopy (XPS) analyses were performed using a PHI Versa Probe II instrument by Physical Electronics, Inc., Chanhassen, MN, USA equipped with a scanning electron Al anode X-ray source (Al_Kα1,2_, E = 1486.6 eV) and a crystal monochromator. A pass energy of 47 eV and a take-off angle of 45° were employed. The binding energy scale was calibrated by referencing the C 1s peak of adventitious hydrocarbons (C-H bonding) at 284.8 eV. Charge neutralization was achieved using simultaneous argon ion and electron flood guns.

Electrical resistance measurements of pure and Sr-doped LaCoO_3_ and LaFeO_3_ thin films in NO_2_ gas were carried out in a 30 cm^3^ chamber. Temperature was measured using a Pt100 sensor and an Agilent 34970A digital multimeter by Keysight Technologies, Santa Rosa, CA, USA. A Keithley 6517 electrometer by Keithley Instruments, part of Tektronix, Beaverton, OR, USA, operating in internal voltage source mode, was used to measure the electrical resistance of the samples. A gas flow control system was used to achieve a specific mixture composition and humidity. Changes in sample resistance were analyzed in the presence of 50 ppm NO_2_ at a total flow rate of 200 cm^3^⋅min^−1^.

## 4. Research Results and Observations

### 4.1. XRD Analysis

Grazing incidence X-ray diffraction analysis was performed at an angle of α = 1°. Phase identification was carried out using JCPDS card numbers as follows: 04-007-6831 (LaCoO_3_), 00-028-1229 (La_0.9_Sr_0.1_CoO_3_), 04-007-8983 (La_0.8_Sr_0.2_CoO_3_), 04-008-0622 (LaFeO_3_), 04-007-6515 (La_0.9_Fe_0.1_CoO_3_), and 00-035-1480 (La_0.8_Fe_0.2_CoO_3_). No other adverse phases, such as La_2_O_3_, CoO, FeO, Co_3_O_4_, Fe_3_O_4_, and Sr(Co,Fe)O_3_, were detected in all samples analyzed (Figure 4). The phase composition of the thin films exhibits a high degree of stability, indicating that the laser ablation process employed (PLD) did not induce any decomposition in the gas phase, as theoretically predicted. This suggests that the material transfer within the plasma cloud occurred in a stoichiometrically congruent manner. These findings strongly reinforce the advantages of utilizing ablation techniques for the deposition of chemically and phase-complex materials.

A (012) preferred orientation was observed in the LaCoO_3_ and La_0.8_Sr_0.2_CoO_3_ films (Figure 4a). In contrast, the La_0.9_Sr_0.1_CoO_3_ film exhibited a dominant (104) orientation. Line broadening slightly marked and observed in the diffraction patterns around 2θ = 55° for all samples is attributed to W L-β emission from the Si substrate, as reported by Tarasov et al. [35]. A slight shift in the diffraction peaks towards higher 2θ values was observed for the Sr-doped lanthanum cobaltite films compared to the undoped LaCoO_3_ (Figure 4a). This shift is attributed to the doping process. Although La^3+^ (1.36 Å) and Sr^2+^ (1.32 Å) ions have similar ionic radii, the substitution of La^3+^ with Sr^2+^ necessitates charge compensation. This leads to an increase in the cobalt oxidation state from Co^3+^ to Co^4+^ and/or the formation of oxygen vacancies. The change in the cobalt ionic radius (6-coordinate, octahedral, and high spin) with oxidation state, from Co^3+^ (0.75 Å) to Co^4+^ (0.67 Å), explains the observed phenomenon of shifting peak positions.

All La(Sr)FeO_3_ thin films showed an unchanged and clearly marked preferred orientation (112) (Figure 4b). A shift in the diffraction peaks towards higher 2θ values was once more observed for Sr-doped samples in comparison to undoped LaFeO_3_. This shift, along with changes in lattice parameters, is attributed to the substitution of La^3+^ ions with Sr^2+^. To maintain charge neutrality in La_x_Sr_1-x_FeO_3_ films, trivalent Fe^3+^ undergoes oxidation to Fe^4+^. This change in the iron oxidation state results in deformation of the unit cell, likely due to the difference in ionic radii between Fe^3+^ (0.65 Å) and Fe^4+^ (0.58 Å). Similar unit cell changes have been observed in Ca-doped LaFeO_3_ materials [36].

### 4.2. XPS Analysis

XPS spectra of the La 3d, Co 2p, and Sr 3d lines for lanthanum cobaltite thin films are shown in Figure 5. The La 3d spectral line (3d_5/2_ and 3d_3/2_) in all samples (Figure 5a,c) consists of two doublets, despite lanthanum being present in only one oxidation state. Both doublets have comparable intensities. The first doublet at a lower binding energy is the main line (BE ≈ 833 eV), while the second at a higher binding energy is the satellite. The presence of these two doublets arises from the complex electron configuration in lanthanum oxides or perovskites (3d_9_f_0_L or 3d_9_f_1_L_−1_, where L denotes the oxygen ligand). The position and intensity ratio of the component doublets indicate that lanthanum has the same oxidation state of +3 in all analyzed samples. Similarly, the Co 2p spectrum in all samples (Figure 5b,d) consists of two doublets (Co 2p_3/2_ and Co 2p_1/2_)—the main line and the satellite, respectively. The stronger doublet at a lower binding energy (BE ≈ 780 eV) is the direct photoelectron line, while the satellite doublet is formed through a shake-up process. The position and shape of the Co 2p spectral lines are nearly identical in both samples, suggesting that cobalt is in the same chemical state. However, determining the specific oxidation state is challenging because the Co 2p line does not exhibit significant shifts in binding energy with changes in Co oxidation state. Based on the position of the Co 2p lines, the oxidation state of cobalt could be +2, +3, or +4. The Sr 3d strontium spectral line, observed only in Figure 5e, consists of two doublets (3d_5/2_ and 3d_3/2_). The doublet with lower binding energy (BE ≈ 132 eV) is attributed to SrO, while the doublet with higher binding energy (BE ≈ 134 eV) corresponds to SrCO_3_. In both compounds, strontium has an oxidation state of Sr^2+^. The presence of strontium carbonate is likely due to the reaction of SrO with incidental carbon, a common surface impurity [37].

The La 3d_5/2_ spectra for all analyzed lanthanum ferrite La(Sr)FeO_3_ films do not change on the cross-section (Figure 6). The spectrum has a main peak at BE ≈ 833 eV and a satellite. The energy values at which the peaks occur are typical for rare earth oxides [38]. The Fe 2p and Sr 3d spectra were analyzed. The Fe 2p_3/2_ and Fe 2p_1/2_ spectra for the respective thin films are shown in Figure 6b,d,e. The Fe 2p_3/2_ spectrum exhibits a main peak at a binding energy (BE) of 706.3 eV and a satellite peak separated by ~4 eV. The Fe 2p_1/2_ peak (BE ≈ 720 eV) also shows a satellite. The obtained binding energies are lower than those typical for Fe-O oxides, where the Fe 2p_3/2_ peak for Fe^3+^ is located in the range of 710 ÷ 711 eV [39]. Such low binding energy of Fe 2p_3/2_ has been observed in LaFeO_3_ perovskites prepared by the sol-gel method. The presence of the satellite peak near the Fe 2p_1/2_ peak indicates the presence of Fe^4+^ [39]. The Sr 3d spectra (Figure 6e) consist of two peaks: Sr 3d_5/2_ (BE ≈ 133 eV) and Sr 3d_3/2_. The position of the Sr 3d_5/2_ peak indicates the presence of Sr^2+^, e.g., in SrO.

### 4.3. Microstructure of Thin Films (SEM Observation and EDS Analysis)

Scanning electron microscopy (SEM) analysis of La(Sr)CoO_3_ and La(Sr)FeO_3_ thin films (Figure 7 and Figure 8) reveals that both stoichiometric perovskite and Sr-doped films exhibit compact structures free of surface defects such as pores or cracks. While occasional particle clusters (aggregates larger than surface irregularities) are observed on the film surfaces, they are typically comparable in size to existing (and equally sporadic) droplets. During the ablation process, particularly at high energy densities or reduced target–substrate distances, larger quantities of target material may be dislodged and transferred to the substrate within the plasma plume. However, the presence of these clusters, if limited, does not significantly impact the film quality or properties, as their chemical composition is nearly identical to the compact, crystalline structure (Table 2 and Table 3). Energy-dispersive X-ray spectroscopy (EDS) analyses confirm that all thin films maintain the desired stoichiometry without over-enrichment of La, Fe, Co, Sr, or O. Although the observed clusters are not detrimental, larger droplet formation can be mitigated through the use of large, polished targets and optimization of the target–substrate distance for the specific deposition atmosphere. Operating at excessively low energy densities necessitates more pulses to achieve the desired coating thickness and can lead to the formation of high cones on the target, which is undesirable. Conversely, excessively high energy densities should be avoided to prevent the generation of sizable aggregates and increased droplet formation on the film surface. Therefore, optimizing process parameters, including photon energy, oxygen partial pressure, target–substrate distance, and substrate temperature, is crucial for minimizing droplet formation and achieving high-quality films. While complete elimination of droplet formation is impossible due to the inherent nature of the pulsed laser deposition (PLD) process, careful parameter control can effectively limit their size and frequency. A key observation in Sr-doped LaCoO_3_ and LaFeO_3_ thin films is the significant refinement of the surface structure (topographical indication). This clearly suggests that Sr doping, in addition to altering the ionic (internal) structure, promotes the formation of smaller crystallites during film growth. SEM observations support the crystallite size analysis performed using XRD (Williamson–Hall analysis results illustrated in Figure 4) and further suggest that these changes in topography should also be observable via atomic force microscopy (AFM).

### 4.4. Topographic Analysis of Thin Film Surfaces (AFM Study and Roughness Parameter Measurements)

Atomic force microscopy (AFM) analysis of the LaCoO_3_ and LaFeO_3_ thin films (Figure 9), conducted using tapping mode in air, confirms the observations made at a larger scale via scanning electron microscopy (SEM) (Figure 7). The AFM images reveal that the film surfaces are composed of fine irregularities, or bulges, which form the thin film itself. Droplets, limited to diameters of less than 1 µm, are observed only in certain areas. Notably, the surface morphologies of the two perovskite films exhibit distinct characteristics. The LaCoO_3_ film displays a uniform and regular surface topography with consistent shapes among the topographic elements. In contrast, the LaFeO_3_ film shows diverse nanoclusters, and its surface can be broadly divided into two regions: one characterized by grains with sharp tips, and the other by grains with flat ends. These nanometer-sized grains, characterized by sharp edges (needle-like morphology), typically exhibit elongation in multiple directions and encircle flatter regions on the thin film surfaces.

The AFM images shown in Figure 10 for Sr-doped LaCoO_3_ and LaFeO_3_ thin films confirm previous observations (SEM topography observations, Figure 7), clearly showing significant surface structure fragmentation as Sr doping progresses. Moreover, LaFeO_3_-based perovskites (Figure 10b) exhibit a distinctly homogenized surface topography. Surface features become more regular and rounded compared to the irregular (dual) surface morphology characteristic of stoichiometric LaFeO_3_ perovskite (Figure 9b). These observations are quantitatively supported by the roughness parameters measured for all (pure and Sr-doped) LaCoO_3_ and LaFeO_3_ perovskites, which are compiled and presented in Table 4.

While in the case of La(Sr)CoO_3_ thin films changes in surface morphology and topography are clearly reflected with a reduction in roughness parameters (Table 4), in the case of doped lanthanum ferrite La(Sr)FeO_3_, apparent surface changes together with Sr doping (reduction in the size of the irregularities and their higher proportion in the analysis area from that of stoichiometric LaFeO_3_) gently raise the roughness. Nevertheless, the roughness values still indicate that the perovskite films are smooth, as the observed changes are on the order of a few nanometers. This topography is characteristic of ablation techniques and has been observed in other functional oxides, such as (Co,Ca)O thin films, prepared by PLD [40].

### 4.5. Nanoscale Hardness and Scratch Testing (Adhesion Measurements)

Nanomechanical properties of thin films were measured according to standardized guidelines for thin film characterization, with particular attention paid to the penetration depth and film thickness. Measurements were performed using a Vickers indenter with a Berkovich diamond tip under a load of 2.5 mN. Example measurement curves for LaFeO_3_ are presented in Figure 11 along with an AFM image of the resulting nanoindentation. It is important to note that, even when standardized guidelines are strictly followed, the measurement result obtained for such thin films should be considered as representative of the specific system comprising the substrate and the thin film deposited on it.

While the measured changes in nanohardness are quite small (Table 5), Sr doping generally appears to increase the hardness of the perovskite thin films across all samples. The only exception lies in the derived Young’s modulus for the film/substrate system. In this case, the slightly reduced value (189 and 191 GPa) observed in the La(Sr)FeO_3_ perovskite, compared to the initial LaFeO_3_ perovskite film (222 GPa), is likely attributed to the surface roughness and topographic elements present in the analyzed areas (SEM and AFM observations previously discussed). Despite this, the nanohardness values for all samples are within the typical range observed for oxide thin films produced by ablative techniques.

Scratch tests were conducted on La(Sr)CoO_3_ and La(Sr)FeO_3_ thin films (Table 5) using a linear progressive scratch (2 mm) with a load ranging from 0.1 to 50 mN at a rate of 5 mN/min. These tests revealed subtle differences in adhesion between pure and Sr-doped perovskite thin films. The critical load (Lc) value for the Si-deposited LaCoO_3_ thin film exceeded 18 mN, while both La_0.9_Sr_0.1_CoO_3_ and La_0.8_Sr_0.2_CoO_3_ exhibited improved adhesion with L_c_ values greater than 24 mN and 21 mN, respectively. These findings correlate with the mechanical properties of the analyzed La_1−x_Sr_x_CoO_3_ thin films, specifically hardness. The addition of Sr into the LaFeO_3_ perovskite structure (critical load, L_c_ > 13 mN) similarly led to improved film adhesion to the Si substrate. This enhancement was observed with critical loads exceeding 19 mN for La_0.9_Sr_0.1_FeO_3_ and 22 mN for La_0.8_Sr_0.2_FeO_3_. Similar to the nanohardness results, the adhesion values obtained in this study fall within the range reported for other oxide thin films deposited [41] using common ablation techniques.

### 4.6. TEM Microstructure Investigation

Cross-sectional transmission electron microscopy (TEM) analysis, utilizing dark and bright field imaging coupled with electron diffraction, revealed that all LaCoO_3_ LaFeO_3_, La(Sr)CoO_3_, and La(Sr)FeO_3_ thin films consisted of nanocrystalline structures with minor local irregularities characteristic of laser ablation deposition (Figure 12, Figure 13, Figure 14 and Figure 15). Both the stoichiometric LaCoO_3_ (Figure 12) and La(Sr)CoO_3_ (Figure 13) perovskite films exhibited columnar crystallites that grew with sharp outlines directly from the Si/MgO substrate interface. Selected area electron diffraction (SAED) patterns confirmed (in line with previous XRD analyses) the presence of the desired phases only: LaCoO_3_ (Figure 12d) and La_0_._9_Sr_0_._1_CoO_3_ (Figure 13d). All La(Sr)CoO_3_ thin films exhibited similar thicknesses, with an average value of 120 ± 5 nm as measured from cross-sectional TEM images (Figure 12 and Figure 13). Based on this, the estimated deposition rate for stoichiometric and Sr-doped LaCoO_3_ thin films was approximately 2.2 × 10^−3^ nm/pulse.

The compact columnar crystal structure was also confirmed for LaFeO_3_ thin films and their Sr-doped variants (Figure 14 and Figure 15). Structure refinement, previously defined on the basis of topographic analyses (SEM, AFM), was confirmed on cross-sections of the deposited lanthanum ferrite perovskites. Importantly, the choice of deposition parameters proved to be accurate in this case as well, as the observed structure is of high quality; the columns are clear, free of deformation and impurities. Such a structure allows us to conclude that the obtained films are structurally within zone I or T, according to the widely accepted Thornton growth model [42]. Visual verification of the thickness (about 120 nm) confirmed the deposition rate previously estimated for La(Sr)CoO_3_ (~2 × 10^−3^ nm/pulse). It is important to highlight that the high-quality columnar internal structure was consistently observed irrespective of the substrate employed (Si or MgO). The only distinction noted was the presence of a thin (nanometric) SiO_2_ oxide film on Si substrates. However, this oxide sublayer did not impede the growth of the crystalline perovskite films.

### 4.7. Electrical Resistance Measurements

Thin films of deposited LaCoO_3_, LaFeO_3_ and their Sr-doped variants were tested in the presence of 50 ppm NO_2_ at temperatures ranging from 200 to 500 °C. The samples were placed in a cell and heated to 300 °C under continuous airflow for at least 3 h prior to the initial measurements. The purpose of this treatment was to remove contaminants from the sample surface and the cell. To stabilize the measurement signal after reaching the target temperature, the sample was held under voltage with airflow until a stable baseline signal (R_air_) was obtained. To determine the effect of current on sensitivity, the response under NO_2_ at 500 °C was measured at different currents: 0.1 nA, 1 nA, and 10 nA (Figure 16). It was observed that sensitivity did not change with varying current (a constant value of 4% was obtained for all the initial measurements taken). The effect of the current was observed only on the noise-to-signal ratio. Noise decreased with increasing current. Further measurements were carried out using 1 nA.

The electrical resistance response (R) of three La(Sr)CoO_3_ thin films to 50 ppm NO_2_ was investigated at various operating temperatures (230 °C, 300 °C, 350 °C, and 440 °C). The results are presented in Figure 17. Prior to gas exposure, the sensor response was stabilized for several hours at a constant temperature of 230 °C. Subsequently, the gas environment was cycled between pure air and 50 ppm NO_2_ while maintaining a constant total gas flow of 200 cm^3^ min^−1^ and a relative humidity of 50% ± 2%. Figure 17 illustrates the decrease in resistance with increasing temperature, a characteristic behavior of extrinsic semiconductors. This can be attributed to the increased number of thermally generated electrons at higher temperatures, leading to enhanced conductivity and a corresponding decrease in resistivity. Furthermore, the introduction of the oxidizing gas (NO_2_) resulted in a decrease in resistance at each measured temperature. This observation is likely due to the adsorption of NO_2_ onto the surface of the p-type LaCoO_3_-based semiconductor, coupled with the oxidizing nature of NO_2_.

The NO_2_ sensing response (Resp) and sensitivity (S) of the thin films were calculated using the following formulas based on experimental data:Resp = R_NO_2__/R_air_,(1)S = ((R_NO_2__ − R_air_)/R_air_) 100%,(2)
where R_NO_2__—stable measured electrical resistance in NO_2_ atmosphere; R_air_—stable measured electrical resistance in air.

The results of these calculations are summarized in Table 6 for La(Sr)CoO_3_ thin films and in Table 7 for La(Sr)FeO_3_ thin films.

The analysis of electrical resistance changes over time for La(Sr)CoO_3_ thin films at various temperatures (Figure 17 and Table 6) reveals several key findings. First, the response (Resp) to both NO_2_ and air remains stable (1.0 ÷ 1.3) across all doping levels. However, response and recovery times (t_res_, t_rec_) are consistently long, at approximately 30 min. While LaCoO_3_ and La_0.9_Sr_0.2_CoO_3_ thin films (Figure 17a,b) exhibit low and unstable sensitivity (S), the La_0.8_Sr_0.2_CoO_3_ thin film (Figure 17c) demonstrates promising characteristics. Despite long response and recovery times, this film shows high stability in the 230 ÷ 440 °C range and increased sensitivity (23 ÷ 30%) up to 350 °C. This suggests that doping with strontium at this level significantly improves the catalytic properties of lanthanum cobaltite (LaCoO_3_) thin films deposited by pulsed laser deposition (PLD).

The measurements of catalytic properties in the presence of NO_2_ for La(Sr)FeO_3_ thin films, particularly for stoichiometric LaFeO_3_, yielded widely varied results. As shown in Figure 18 and Table 7, significant variations in response (Resp = 0.3 ÷ 2.7) and sensitivity (S = 75 ÷ 3.8%) were observed across the analyzed temperature range (230 ÷440 °C). Response and recovery times (t_res_ = 25 ÷ 5 min, t_rec_ = 14 ÷ 5 min) also varied, though they were generally shorter. Most notably, in the LaFeO_3_ thin film (Figure 18), a shift in conductivity type was observed between 350 and 400 °C. Initially, the film exhibited typical n-type semiconductor behavior. However, as the temperature increased at a constant NO_2_ flow rate and concentration, the conductivity transitioned to p-type. Certain materials exhibit a unique property: their electrical conductivity type can change in response to variations in temperature or gas pressure. As demonstrated by Minh and Takahashi [43], increasing the partial pressure of oxygen can induce a shift in the dominant charge carrier. These materials exhibit three distinct conductivity zones: n-type, mixed, and p-type. The mixed conductivity zone, characterized by the simultaneous presence of both n-type and p-type conductivity, exhibits lower overall conductivity compared to the purely n-type or p-type zones. As oxygen pressure increases, conductivity initially decreases, leading to the emergence of the mixed conductivity region. This region, marked by competing conductivity mechanisms, displays the lowest electrical conductivity. Further increases in oxygen pressure ultimately cause a transition to p-type conductivity, where conductivity increases proportionally with pressure. This behavior highlights that resistive gas sensors possess an optimal operating point defined not only by temperature, but also by gas pressure.

Data collected for two Sr-doped LaFeO_3_ thin films (Figure 19 and Figure 20 and Table 7) do not exhibit the drastic conductivity changes observed previously, further suggesting that Sr doping stabilizes the catalytic activity of this material. These films demonstrated relatively fast response and recovery times (15 min) and stable response (Resp = 1.0 ÷ 1.2). Sensitivity ranged from S = 4.8 ÷ 8.6% for La_0.9_Sr_0.1_FeO_3_ (Figure 19) to S = 9.7 ÷ 17.9% for La_0.8_Sr_0.2_FeO_3_ (Figure 20), both measured between 230 and 350 °C. The final measurements (Figure 19d and Figure 20d) were significantly affected by a low signal-to-noise ratio. In these samples, noise increased with temperature, rendering the final measurement for 400 ÷ 440 °C completely unreadable. This made it difficult to determine individual values for response, sensitivity, and recovery/response times.

Variations in the reaction and regeneration times of Sr-doped LaCoO_3_ and LaFeO_3_ perovskite thin films arise from a complex interplay between the material’s structure (surface), adsorption–desorption mechanisms, and the reaction kinetics with the active gas. Strontium doping significantly alters these properties by generating oxygen vacancies (with concentrations closely correlated to strontium content) and modifying the cation oxidation states. Given a structural analysis of the synthesized perovskite thin films, and assuming a consistent interaction mechanism of surface oxygen defects at the active sites across all samples, the disparities in response and regeneration times can be attributed to surface topography. The observed structural fragmentation in La(Sr)FeO_3_ compared to La(Sr)CoO_3_ (as evidenced by SEM and AFM images, and measured roughness parameters) appears to correspond to a reduction in response and regeneration times. Decreasing crystallite size increases the concentration of active centers, thereby substantially stabilizing the sensitivity of thin films. As demonstrated, optimizing the structure of the gas-sensitive material is essential for maintaining high catalytic performance.

## 5. Summary and Discussion

In addition to the perovskites discussed in this paper, several other thin-film materials are being intensively studied for use in gas sensors, including NO_2_. These primarily include simple metal oxides (SnO_2_, TiO_2_, In_2_O_3_, and WO_3_ [44,45,46,47]), known for their high sensitivity to various gases, including NO_2_. They can be fabricated into thin films with controlled structures using various techniques, allowing for the adjustment of their catalytic properties. Graphene, molybdenum disulfide, and other transition metal halides are also promising. These materials, due to their large specific surface area and unique electronic properties, show potential in gas detection.

Perovskites, including La(Sr)CoO_3_ and La(Sr)FeO_3_, which are the subject of this research, often exhibit improved sensitivity to a broad range of gases, including volatile organic compounds (VOCs), nitrogen oxides, and carbon monoxide. Their crystal structure and surface properties facilitate strong interactions with gas molecules, leading to significant changes in electrical conductivity. In contrast, traditional metal oxides may have limited sensitivity to certain gases or require higher operating temperatures, which often restricts their application in sensors. Furthermore, by appropriately adjusting the perovskite composition (precise doping), its selectivity towards specific gases can be tuned. This tunability stems from the ability to alter the oxygen binding energy and create oxygen vacancies in the crystal lattice, which selectively interact with specific gas molecules. Achieving high selectivity with traditional metal oxides can be challenging due to their lower compositional and structural flexibility.

Perovskite-based sensors can operate at lower temperatures compared to traditional oxide sensors. This advantage reduces energy consumption and improves sensor lifespan. Traditional metal oxides often require high temperatures to achieve optimal performance, leading to energy consumption and stability issues. Perovskites often exhibit faster response times to changes in gas concentration. This rapid response is crucial in applications requiring real-time monitoring. Traditional oxide sensors may have slower response times, limiting their usefulness in high-dynamic-range applications. The perovskite structure allows for a vast variety of compositions by incorporating different metals and non-metals. This diversity enables the tailoring of sensor properties to meet the requirements of specific applications. Traditional metal oxides tend to have more fixed compositions, offering less flexibility in sensor design.

Modeling the electrical resistance of La(Sr)CoO_3_ and La(Sr)FeO_3_ perovskite thin films with varying strontium (Sr) doping concentrations presents a complex challenge. A critical consideration involves changes in the ionic structure, particularly the creation of oxygen vacancies. These alterations can compromise the integrity of the base perovskite, even at low Sr doping levels (a few atomic percentages). In this study, we deliberately selected 10% and 20% Sr doping to produce targets with precise stoichiometric compositions: La_0.9_Sr_0.1_CoO_3_, La_0.8_Sr_0.2_CoO_3_, La_0.9_Sr_0.1_FeO_3_, and La_0.8_Sr_0.2_FeO_3_ as proven by the material analysis presented above (XRD, SEM/EDS, TEM, AFM, and XPS).

Initially, obtaining significant and reproducible resistance (conductivity) changes at low dopant concentrations (1 ÷ 5% Sr) proves difficult, as substrate influence can obscure measurements. Conversely, excessive Sr doping (approaching 40 ÷ 50% Sr and above) induces substantial distortions in the cationic sublattice and increases the risk of surface defects [48]. These defects can generate unfavorable tensile stresses, outweighing the benefits of increased surface chemically active centers (relevant to catalysis). Consequently, measurements become dominated by structural changes, introducing significant errors. The chosen Sr doping concentrations define a range for substantial conductivity (resistivity in the presence of NO_2_) variations, enabling a planned, progressive narrowing of these ranges in subsequent studies.

Increased strontium concentration influences charge carrier hopping between cobalt (Co) and iron (Fe) ions of varying valences, thus modifying resistance (as discussed in XRD studies). At higher temperatures (above 500 °C), polaron transport may become dominant, especially in highly defective materials (with defect concentrations up to several tens of percent). These observations suggest an optimal strontium doping range of a few to approximately 15%, which balances high conductivity with structural compactness, homogeneity, and controlled surface roughness. Exceeding or falling below this range can lead to uncontrolled increases in resistance.

External factors, notably temperature, significantly influence La(Sr)CoO_3_ and La(Sr)FeO_3_ electrical resistance. Our modeling incorporates this dependence, hence the 230 ÷ 440 °C measurement range. While limited, this range aligns with the observed reproducible changes in catalytic properties upon exposure to specific external agents. This consideration is crucial for potential applications in resistive sensors, where size and weight constraints necessitate lightweight, chemically resistant support materials (e.g., light non-ferrous metal alloys or polymer composites). This practical requirement restricts the operational temperature to below 500 °C, justifying our chosen range.

This research focuses on verifying real-world film growth conditions, aiming for idealized structures that correlate with predicted catalytic property changes in the presence of NO_2_ within the specified temperature range, rather than predictive modeling of structural and property variations. To optimize the electrical properties of La(Sr)CoO_3_ and La(Sr)FeO_3_ thin films for electronic and electrochemical device applications, a strong understanding of these factors is required.

A key issue, especially in the context of the long-term use of Sr-doped LaCoO_3_ and LaFeO_3_ thin films as gas sensors, is their degradation and changes in catalytic properties occurring during exposure. The degradation process is complex and depends on many factors, some of which can be relatively easily eliminated, while others require more rigorous solutions. The main factors include environmental conditions such as temperature, atmosphere, and thermal cycling. High temperatures, often exceeding the recommended operating range of the material, accelerate degradation processes including ion diffusion, phase changes, and oxidation. Moisture, CO_2_, SOₓ, and other corrosive gases can react with perovskites, leading to adverse and unpredictable changes in their composition and structure. Repeated temperature changes can generate mechanical stresses, initiating cracking and delamination of the layers.

Introducing Sr doping into LaCoO_3_ and LaFeO_3_ affects structural stability and allows control of oxygen ion mobility by optimizing dopant concentration, which translates to the generation of oxygen vacancies responsible for chemical activity. Thinner layers are unfortunately more susceptible to degradation due to the larger contact surface with the environment. Therefore, ensuring high-quality internal structure, i.e., a dense, oriented columnar crystal structure free of defects and discontinuities, achieved through controlled epitaxial growth, is crucial. Such conditions can be achieved using the detailed pulsed laser deposition (PLD) process, as confirmed by material research results (XRD, XPS, SEM, TEM, and AFM).

The degradation rate of gas-sensitive thin films also depends on the nature of catalytic reactions. In some cases, contaminants may deposit on the perovskite surface, blocking active sites and leading to catalyst deactivation. Typical degradation processes include ion diffusion, reactions with gases, phase changes, sintering, and poisoning. At high temperatures, oxygen and metal ions can diffuse within the layer or to its surface, leading to changes in chemical composition and defect formation. Perovskites, despite their thermal stability, can react with moisture, CO_2_, or other gases, forming carbonates, hydroxides, or other compounds that alter their structure and properties. Chemical substances from reaction gases can bind to active sites on the material surface, poisoning the gas-sensitive layer. Excessively high temperatures can cause grain sintering, especially with significant surface porosity, which reduces the active surface area of the catalyst.

These phenomena often manifest as a significant decrease in catalytic activity, as well as a deterioration of selectivity and stability, which shortens the layer’s lifespan. To effectively prevent or limit the degradation of La(Sr)CoO_3_ and La(Sr)FeO_3_ gas-sensitive thin films, it is necessary to optimize the chemical composition and structure of the perovskites, control environmental conditions during catalytic reactions (which is difficult to achieve), and use appropriate dopants that improve structural stability.

In light of the above, the material analyses and structural observations conducted in this work, leading to the determination of optimal PLD process parameters for producing high-quality La(Sr)CoO_3_ and La(Sr)FeO_3_ thin films, are extremely valuable.

Perovskite sensors utilizing pure and Sr-doped LaCoO_3_ and LaFeO_3_ thin films demonstrate significant potential across diverse applications, particularly in gas sensing (detecting NO_x_, CO, VOCs, and other gases) and environmental monitoring (air quality assessment, industrial safety systems, and medical diagnostics). Strontium doping enhances the conductivity and reactivity of these materials, leading to improved sensitivity and selectivity. Furthermore, the temperature-dependent electrical resistance of these films enables their application in temperature sensors, while their sensitivity to humidity variations opens avenues for humidity sensors used in environmental monitoring.

These LaCoO_3_ and LaFeO_3_ perovskite thin films can be fabricated through various methods, including magnetron sputtering, atomic layer deposition (ALD), molecular beam epitaxy (MBE), and pulsed laser deposition (PLD) or pulsed electron deposition (PED), both of which are PVD-based ablation techniques. Ablation techniques, unlike others, minimize gas phase decomposition and, when optimized, ensure precise stoichiometry and chemical/phase composition transfer from target to substrate. Ablation methods, offering precise thickness control through adjustments in parameters such as power density, working distance, and pulse count, are particularly well-suited for producing La(Sr)CoO_3_ and La(Sr)FeO_3_ thin films. This precision facilitates seamless integration into microelectromechanical systems (MEMS), enabling miniaturization and integration with other electronic components.

These fabrication techniques are scalable and relatively cost-effective, promoting the mass production of thin films. Ongoing advancements in ablation, ALD, and MBE technologies allow for highly accurate control of layer thickness and composition, crucial for applications demanding high precision and dimensional repeatability. The low cost of starting perovskite materials further enhances the economic viability of mass production. Moreover, the development of fabrication techniques on flexible substrates paves the way for wearable sensor devices, integrating microelectronic sensors into everyday apparel.

## 6. Conclusions

This study investigated the structural, morphological, and electrical properties of pure and Sr-doped LaCoO_3_ and LaFeO_3_ thin films deposited by pulsed laser deposition (PLD) for potential application as sensing electrodes (SE) in NO_2_ gas sensors at temperatures ranging from 230 to 440 °C. The key findings are as follows:Successful thin film deposition: High-quality, nanocrystalline thin films of LaCoO_3_, LaFeO_3_, and their Sr-doped variants were successfully deposited on Si and MgO substrates using PLD. The films exhibited a compact, columnar structure with minimal defects, confirming the suitability of PLD for depositing complex perovskite materials.Sr doping effects: Sr doping significantly influenced the microstructure and surface topography of the films. It led to a refinement of the surface structure, promoting the formation of smaller crystallites and a more homogenous surface topography, particularly in LaFeO_3_ films. This was confirmed by XRD, SEM, and AFM analyses.Enhanced mechanical properties: Sr doping generally improved the nanohardness and adhesion of the perovskite thin films to the Si substrate, suggesting potential benefits for sensor durability and stability.Electrical response to NO_2_: LaCoO_3_ and LaFeO_3_ thin films exhibited distinct electrical responses to NO_2_ gas. Sr-doped LaCoO_3_, particularly La_0_._8_Sr_0_._2_CoO_3_, showed promising sensing characteristics with high stability and increased sensitivity to NO_2_ at temperatures up to 350 °C. However, long response and recovery times were observed.Conductivity type transition in LaFeO_3_: Stoichiometric LaFeO_3_ exhibited a unique transition from n-type to p-type conductivity with increasing temperature in the presence of NO_2_ (50 ppm). This phenomenon, attributed to changes in the dominant charge carrier under varying oxygen partial pressure, highlights the importance of optimizing both temperature and gas pressure for resistive gas sensors.Sr doping stabilizes LaFeO_3_ response: Sr doping stabilized the catalytic activity of LaFeO_3_, resulting in faster response and recovery times and a more consistent response to NO_2_. However, increased noise at higher temperatures limited the sensitivity measurements.

Overall, this study demonstrates the potential of Sr-doped LaCoO_3_ and LaFeO_3_ thin films as sensing electrodes for NO_2_ gas sensors. Further research is needed to optimize the sensor performance, particularly in terms of response and recovery times, and to investigate the long-term stability and selectivity of these materials in various gas environments. The laser ablation technique (PLD) has been successfully employed to fabricate high-quality, nanocrystalline perovskite thin films for gas sensing, suggesting that this method can be extended to other functional oxide materials.

## Figures and Tables

**Figure 1 materials-18-01175-f001:**
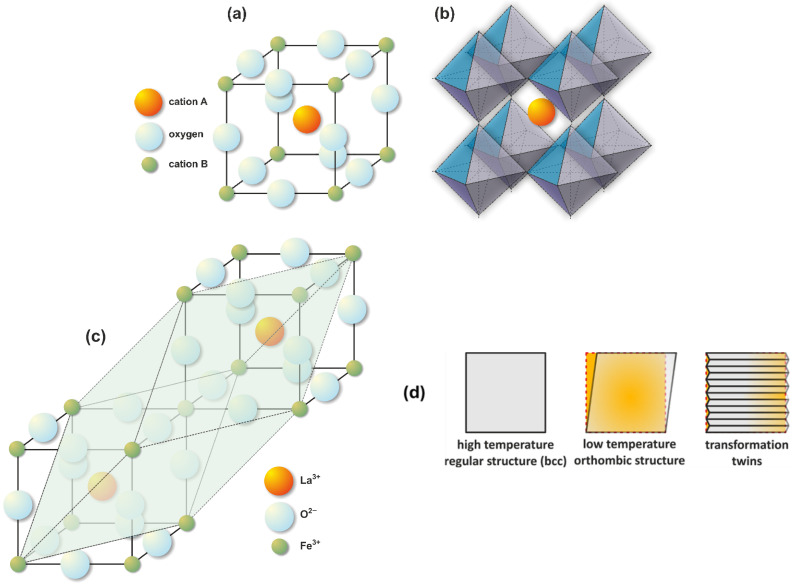
A representative example of the ABX_3_ perovskite structure (**a**), along with its characteristic symmetry (**b**). The structural transformation scheme for LaMO_3_ (M = Co, Fe) as a function of temperature is illustrated in (**c**). This transformation often leads to the formation of twin domains within the material, as depicted in (**d**).

**Figure 2 materials-18-01175-f002:**
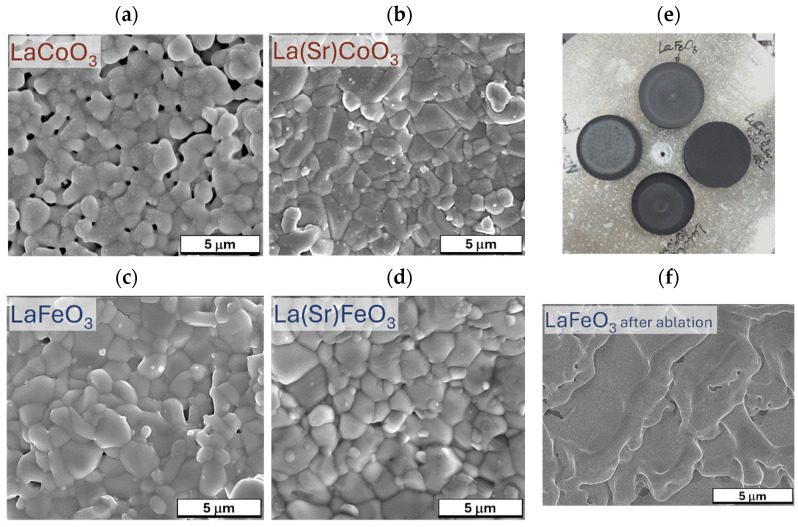
SEM images illustrating the surface topography of the targets used in the study; (**a**) LaCoO_3_, (**b**) La(Sr)CoO_3_, (**c**) LaFeO_3_, (**d**) La(Sr)FeO_3_; and (**e**) macro image of targets ready for microscopic examination and (**f**) the surface of the LaFeO_3_ target after laser ablation.

**Figure 3 materials-18-01175-f003:**
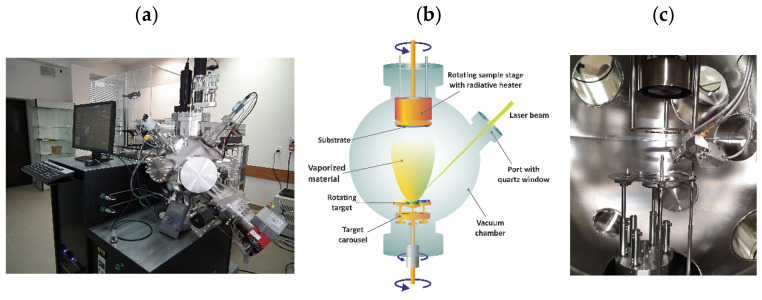
Laser ablation system (PLD) built with an Nd:YAG laser and the Neocera vacuum chamber, connected by an optical system (**a**). Schematic of the PLD process (**b**) and a view inside the process chamber (**c**).

**Figure 4 materials-18-01175-f004:**
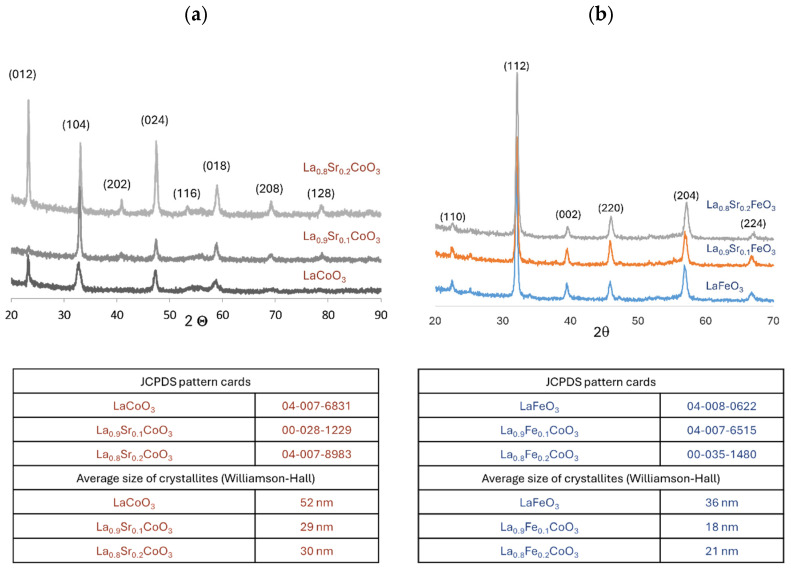
XRD phase analysis of (**a**) Sr-doped LaCoO_3_ and (**b**) Sr-doped LaFeO_3_ thin films, with JCPDS pattern cards and average crystallite sizes (estimated using the Williamson–Hall method).

**Figure 5 materials-18-01175-f005:**
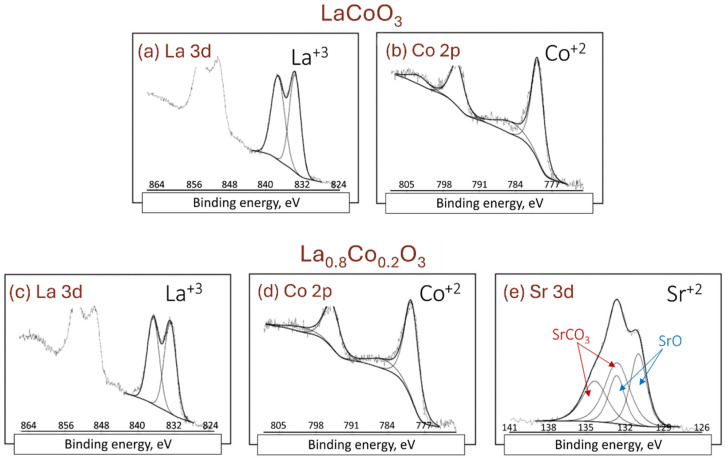
XPS detection/verification of chemical states of elements for La(Sr)CoO_3_ thin films.

**Figure 6 materials-18-01175-f006:**
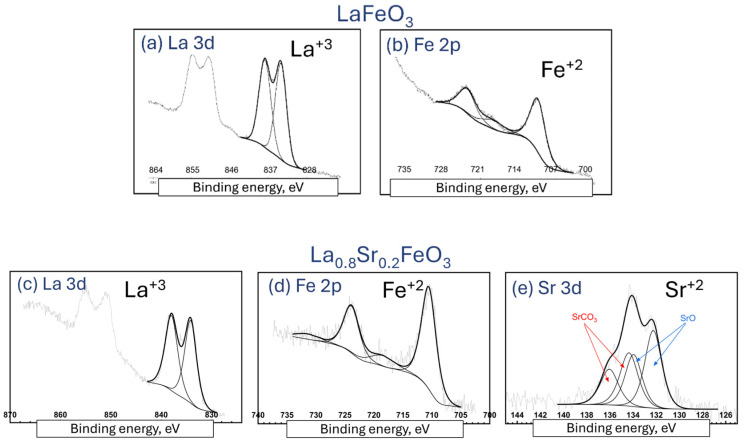
XPS detection/verification of chemical states of elements for La(Sr)FeO_3_ thin films.

**Figure 7 materials-18-01175-f007:**
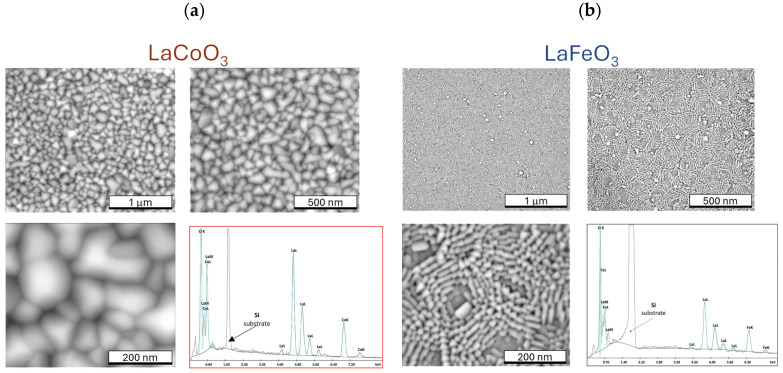
SEM images of the topography of perovskite thin films grown on monocrystalline Si substrates [001] with the result of EDS analysis of the chemical composition for (**a**) LaCoO_3_ and (**b**) LaFeO_3_.

**Figure 8 materials-18-01175-f008:**
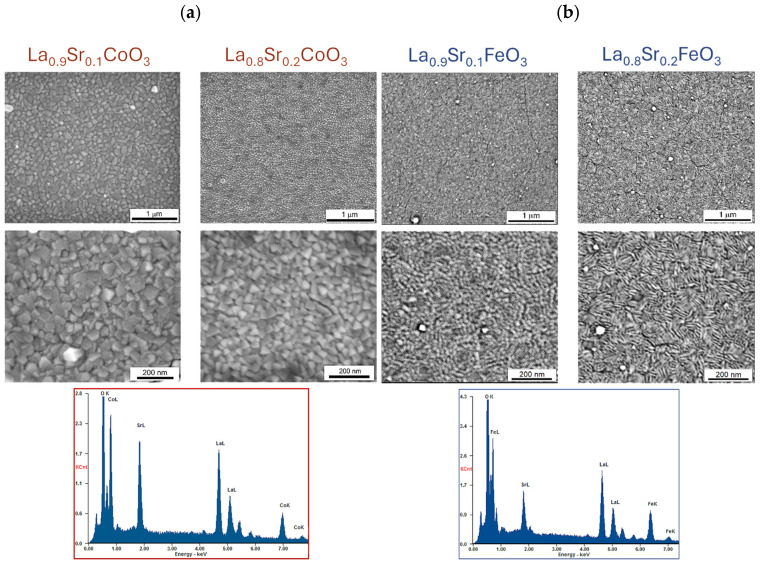
SEM images of the topography of perovskite thin films grown on monocrystalline Si substrates [001] with the result of EDS analysis of the chemical composition for (**a**) Sr-doped LaCoO_3_, (**b**) Sr-doped LaFeO_3_.

**Figure 9 materials-18-01175-f009:**
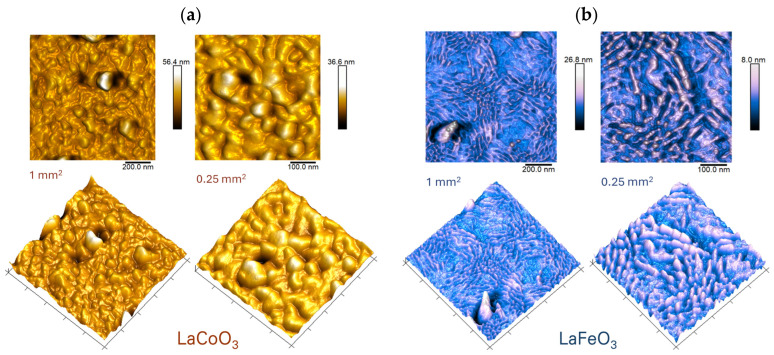
Surface topography images of perovskite thin films by atomic force microscopy (AFM) technique for (**a**) LaCoO_3_ and (**b**) LaFeO_3_.

**Figure 10 materials-18-01175-f010:**
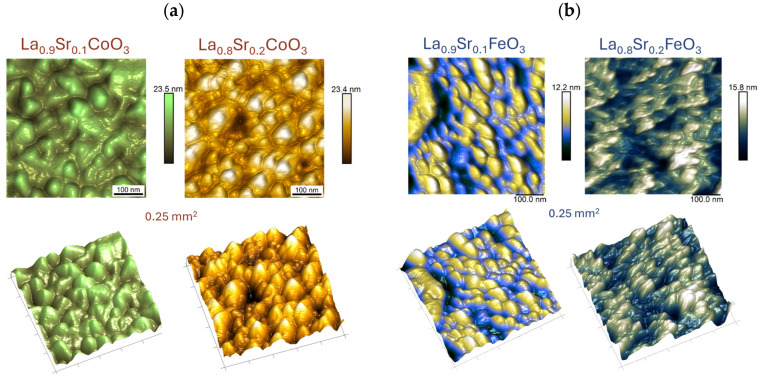
Surface topography images of perovskite thin films by atomic force microscopy (AFM) technique for (**a**) Sr-doped LaCoO_3_ and (**b**) Sr-doped LaFeO_3_.

**Figure 11 materials-18-01175-f011:**
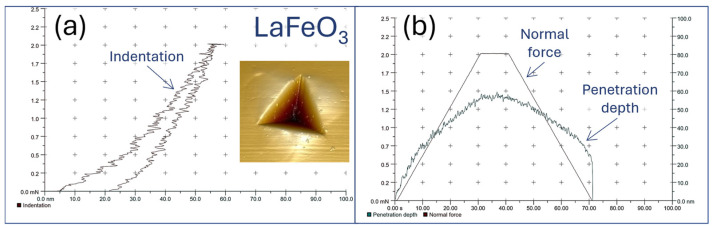
Examples of (**a**) indentation curve and (**b**) penetration depth and normal force plots obtained from nanohardness measurements of LaFeO_3_ thin film on Si substrate.

**Figure 12 materials-18-01175-f012:**
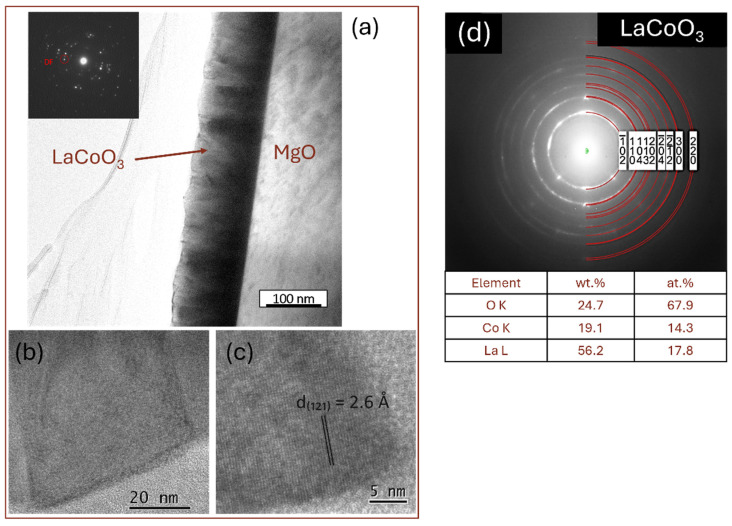
TEM analysis of LaCoO_3_ thin films: (**a**) low-magnification bright-field image with selected area electron diffraction pattern, (**b**,**c**) high-resolution TEM images, and (**d**) SAED solved with TEM/EDS analysis.

**Figure 13 materials-18-01175-f013:**
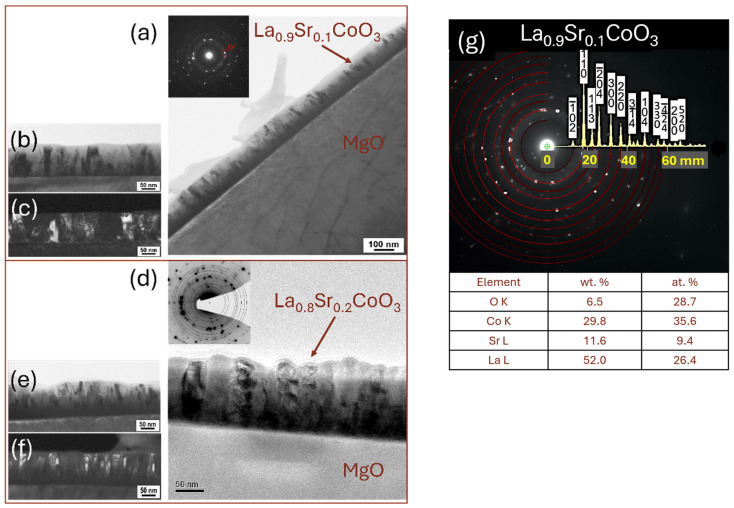
TEM analysis of Sr-doped LaCoO_3_ thin films: (**a**,**b**,**d**,**e**) low-magnification bright- and (**c**,**f**) dark-field image with selected area electron diffraction patterns, and (**g**) SAED solved with TEM/EDS analysis.

**Figure 14 materials-18-01175-f014:**
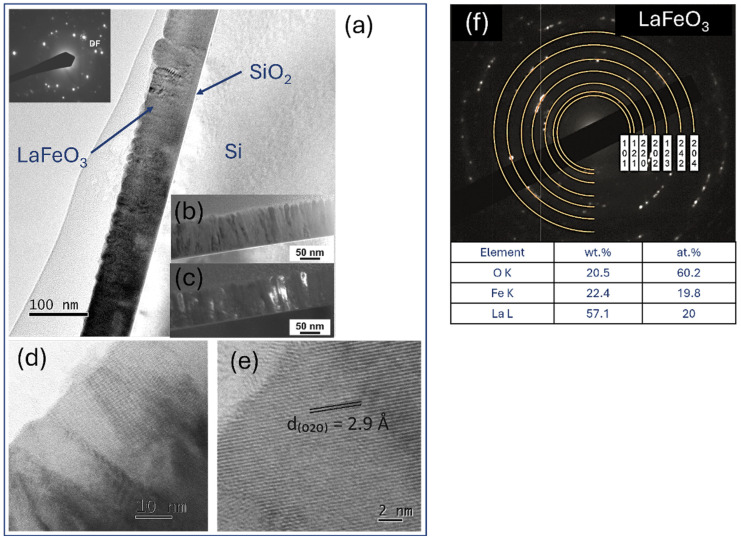
TEM analysis of LaFeO_3_ thin films: (**a**,**b**) low-magnification bright- and (**c**) dark-field image with selected area electron diffraction pattern, (**d**,**e**) high-resolution TEM images, and (**f**) SAED solved with TEM/EDS analysis.

**Figure 15 materials-18-01175-f015:**
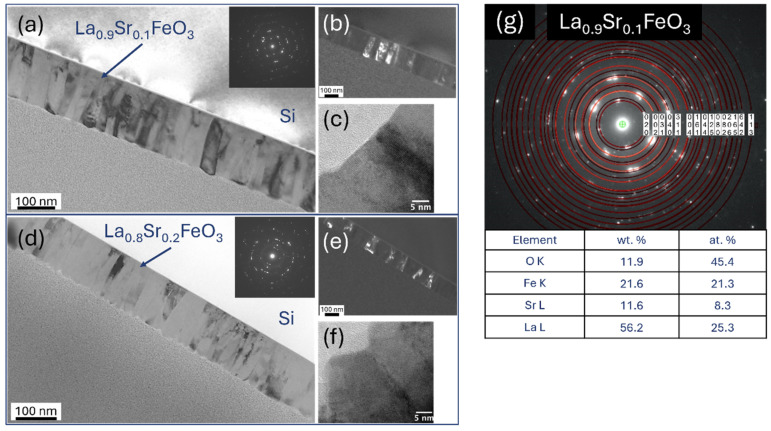
TEM analysis of Sr-doped LaFeO_3_ thin films: (**a**,**d**) low-magnification bright- and (**b**,**e**) dark-field image with selected area electron diffraction patterns, (**c**,**f**) high-resolution TEM images and (**g**) SAED pattern solved with TEM/EDS analysis.

**Figure 16 materials-18-01175-f016:**
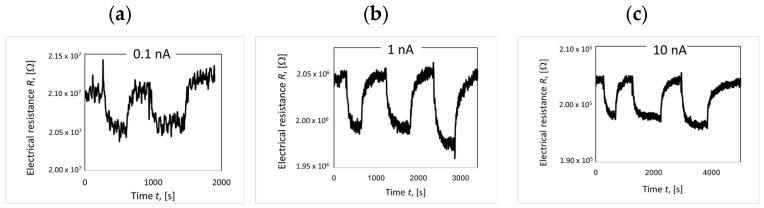
LaCoO_3_ response at 500 °C exposed to 50 ppm of NO_2_ using different currents: (**a**) 0.1 nA, (**b**) 1 nA, and (**c**) 10nA.

**Figure 17 materials-18-01175-f017:**
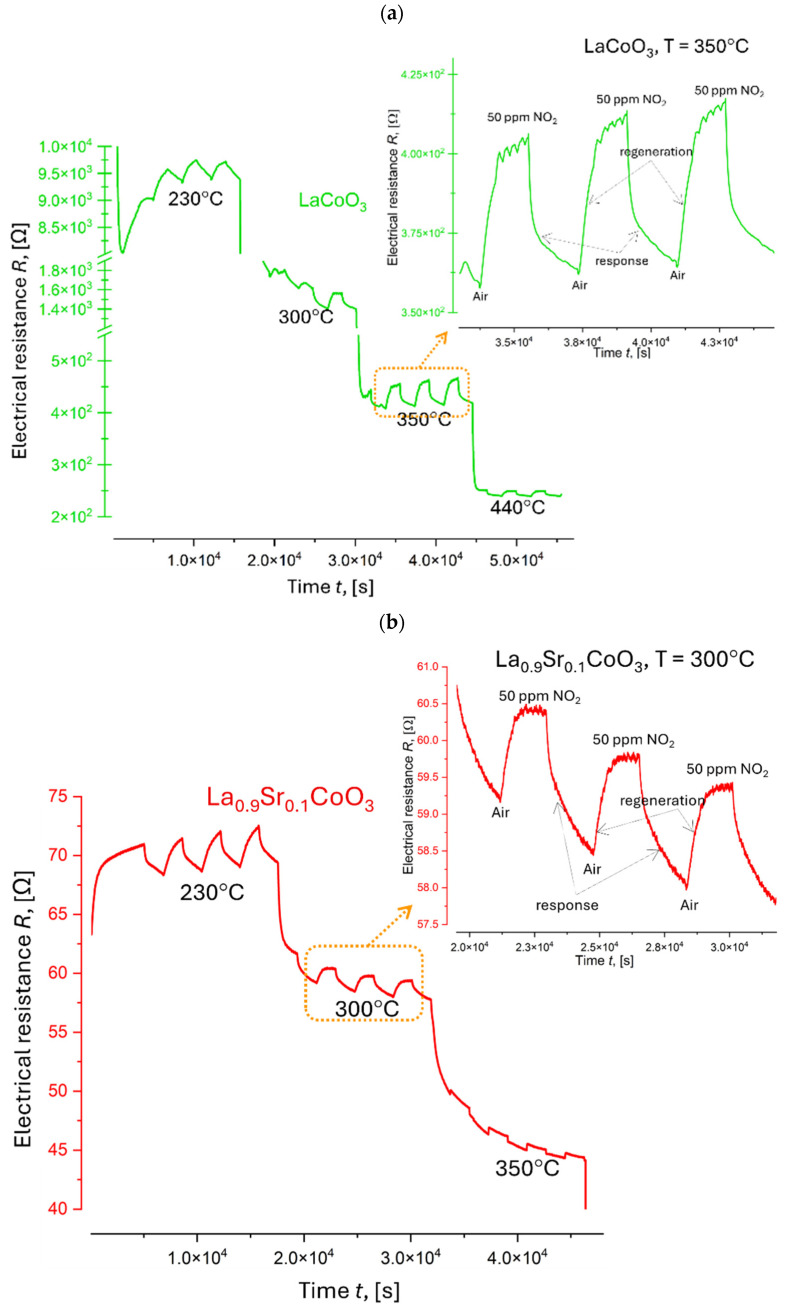
Response of La(Sr)CoO_3_ exposed to 50 ppm NO_2_ at temperatures in the range of 230 ÷ 440 °C: (**a**) LaCoO_3_, (**b**) La_0.9_Sr_0.1_CoO_3_, and (**c**) La_0_._9_Sr_0.1_CoO_3_.

**Figure 18 materials-18-01175-f018:**
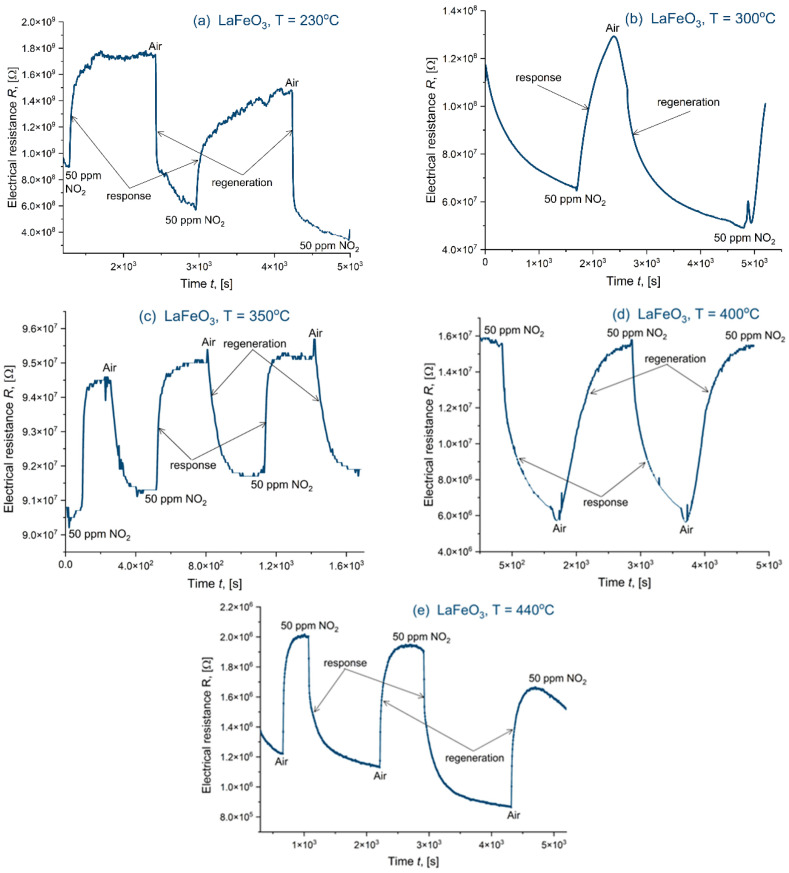
Response of LaFeO_3_ exposed to 50 ppm NO_2_ for a range of temperatures: (**a**) 230 °C, (**b**) 300 °C, (**c**) 350 °C, (**d**) 400 °C, and (**e**) 440 °C.

**Figure 19 materials-18-01175-f019:**
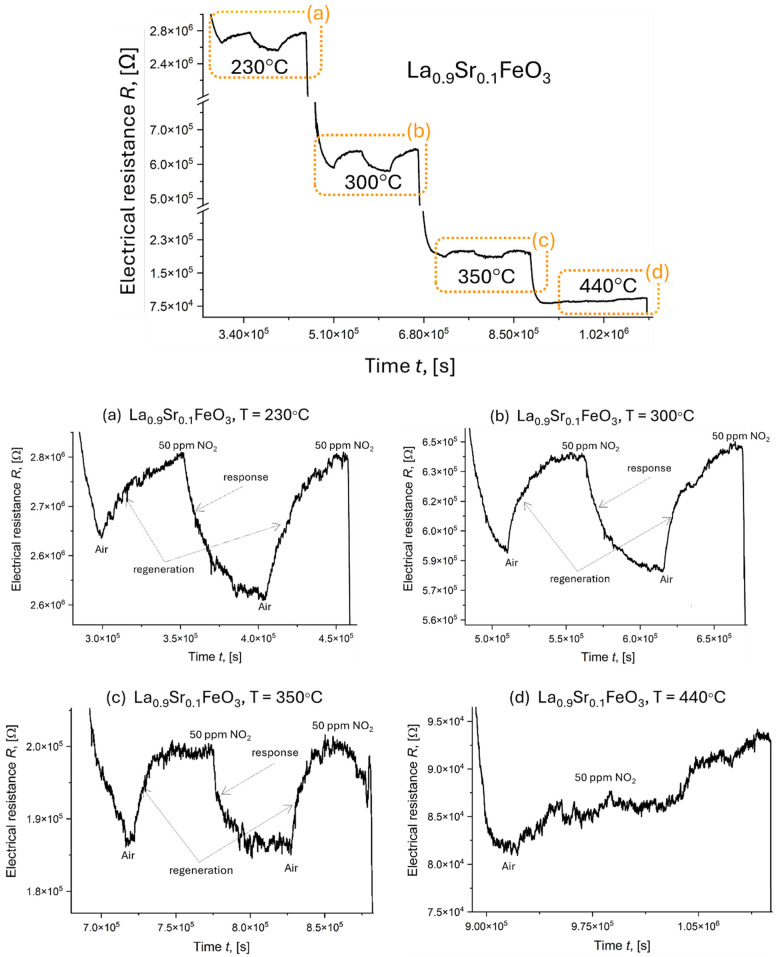
Response of La_0_._9_Sr_0_._1_FeO_3_ exposed to 50 ppm NO_2_ (summary chart) and for a range of temperatures: (**a**) 230 °C, (**b**) 300 °C, (**c**) 350 °C, and (**d**) 440 °C.

**Figure 20 materials-18-01175-f020:**
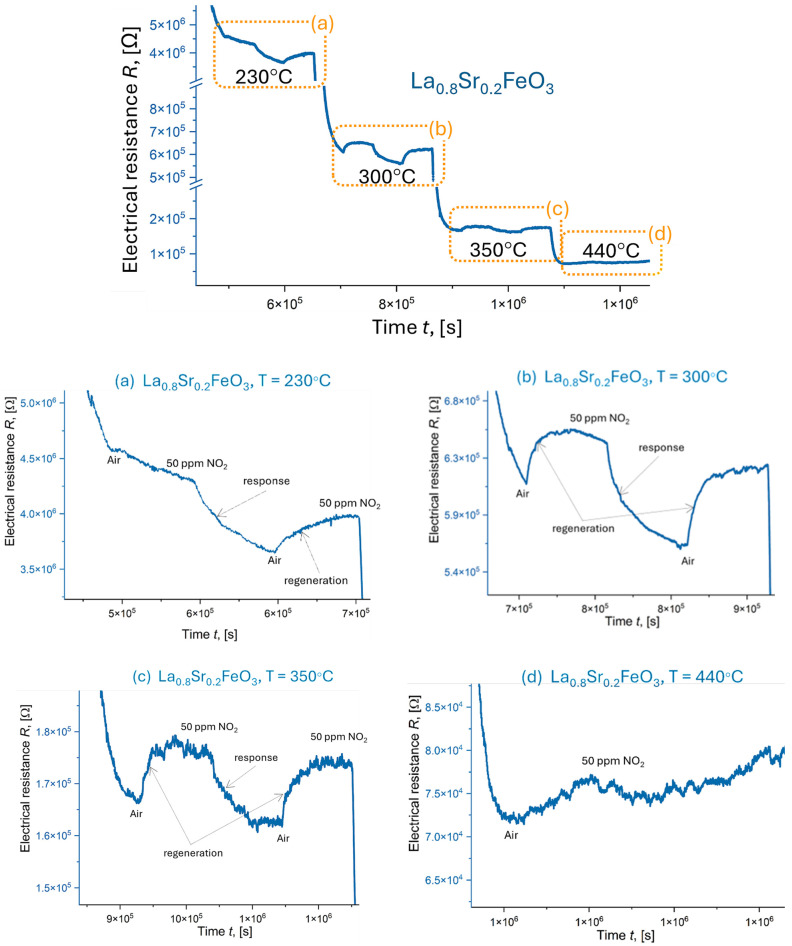
Response of La_0_._8_Sr_0_._2_FeO_3_ exposed to 50 ppm NO_2_ (summary chart) and for a range of temperatures: (**a**) 230 °C, (**b**) 300 °C, (**c**) 350 °C, and (**d**) 440 °C.

**Table 1 materials-18-01175-t001:** Deposition process parameters of La(Sr)CoO_3_ and La(Sr)FeO_3_ thin films.

Process Parameter	Value
Laser wavelength, λ (nm)	266 (IV H)
Pulse Energy, E (mJ)	~90
Energy density, (J·cm^−2^)	~2.0
Target–substrate, d (mm)	~50
Oxygen partial pressure, p_O2_ (Pa)	5.3
Substrate temperature, T_s_ (°C) (radiative heater)	~750
Repetition rate, f (Hz)	10
Number of shots	70,000
Pulse deposition ratio (Å/pulse)	~0.02
Substrates	MgO [001] Si [001]

**Table 2 materials-18-01175-t002:** Results of EDS chemical composition analyses of micro-area La(Sr)CoO_3_ thin films. Correction method: PhiRhoZ. Type: OxyByDiff—oxygen content was estimated using stoichiometry.

Thin Film	La				Sr				Co				O			
	Thin Film		Droplets, Particles		Thin Film		Droplets, Particles		Thin Film		Droplets, Particles		Thin Film		Droplets, Particles	
	wt.%	at.%	wt.%	at.%	wt.%	at.%	wt.%	at.%	wt.%	at.%	wt.%	at.%	wt.%	at.%	wt.%	at.%
LaCoO_3_(stech.)	56.5	20							24	20			19.5	60		
LaCoO_3_	56.3	19.7	58.7	21.0	-	-	-	-	23.6	18.3	21.7	18.2	20.1	62.0	19.6	60.8
La_0.9_Sr_0.1_CoO_3_	42.1	12.7	45.4	15.1	9.8	4.1	10.3	4.7	19.7	13.1	20.3	15.7	28.4	70.1	24.0	64.5
La_0.8_Sr_0.2_CoO_3_	29.5	7.6	32.0	9.0	20.9	8.5	22.2	9.9	16.9	10.3	17.3	11.5	32.7	73.6	28.5	69.6

**Table 3 materials-18-01175-t003:** Results of EDS chemical composition analyses of micro-area La(Sr)FeO_3_ thin films. Correction method: PhiRhoZ. Type: OxyByDiff—oxygen content was estimated using stoichiometry.

Thin Film	La				Sr				Fe				O			
	Thin Film		Droplets, Particles		Thin Film		Droplets, Particles		Thin Film		Droplets, Particles		Thin Film		Droplets, Particles	
	wt.%	at.%	wt.%	at.%	wt.%	at.%	wt.%	at.%	wt.%	at.%	wt.%	at.%	wt.%	at.%	wt.%	at.%
LaFeO_3_ (stech.)	57.2	20	-	-	-	-	-	-	23	20	-	-	19.8	60	-	-
LaFeO_3_	57.4	19	63.3	25.5	-	-	-	-	20	16.4	21.6	21.6	22.6	64.6	15.1	52.9
La_0.9_Sr_0.1_FeO_3_	55.1	24.2	53.5	23.0	11.1	7.7	11.8	8.1	22.0	22.7	22.3	22.6	11.9	45.4	12.4	46.4
La_0.8_Sr_0.2_FeO_3_	40.4	13.9	33.5	10.5	22.3	12.2	25.8	12.8	17.3	14.0	17.3	12.8	20.0	59.9	23.5	63.9

**Table 4 materials-18-01175-t004:** Roughness parameters of La(Sr)CoO_3_ and La(Sr)FeO_3_ thin films.

Sample/Thin Film	Roughness Parameter, (nm)			
	Rq	Ra	Rz	Rmax
LaCoO_3_	2.4	1.8	14.7	20.7
La_0.9_Co_0.1_O_3_	2.6	2.1	17.5	17.8
La_0.8_Co_0.2_O_3_	3.2	2.6	18.7	21.4
LaFeO_3_	0.9	0.7	5.6	6.3
La_0.9_Fe_0.1_O_3_	1.7	1.4	8.6	13.1
La_0.8_Fe_0.2_O_3_	2.3	1.8	11.9	17.0

**Table 5 materials-18-01175-t005:** Selected mechanical properties and adhesion results obtained for La(Sr)CoO_3_ and La(Sr)FeO_3_ thin films.

Thin Film/Substrate	Nanohardness HV	Indentation Hardness H_IT_, (GPa)	Young’s Modulus E_IT_, (GPa)	Penetration Depth h_m_, (nm)	Critical Load L_c_, (mN)
LaCoO_3_/Si	1538 ± 4	16.6 ± 0.2	184 ± 2	61 ± 3	>18 ± 1.0
La_0.9_Sr_0.1_CoO_3_/Si	2382 ± 7	25.7 ± 0.3	196 ± 3	55 ± 2	>24 ± 0.7
La_0.8_Sr_0.2_CoO_3_/Si	2157 ± 5	23.9 ± 0.3	191 ± 3	57 ± 2	>21 ± 0.8
LaFeO_3_/Si	1233 ± 3	13.3 ± 0.1	222 ± 4	60 ± 3	>13 ± 1.1
La_0.9_Sr_0.1_FeO_3_/Si	1798 ± 7	18.4 ± 0.2	189 ± 2	58 ± 2	>19 ± 1.0
La_0.8_Sr_0.2_FeO_3_/Si	2291 ± 8	24.8 ± 0.2	191 ± 3	56 ± 2	>22 ± 0.8

**Table 6 materials-18-01175-t006:** La(Sr)CoO_3_ thin film NO_2_-sensing response (Resp), sensitivity (S), response (t_res_), and recovery (t_rec_) times measured for 230 ÷ 440 °C.

Thin Film	Temperature T [°C]	NO_2_ Sensing Response, Resp	Thin Film Sensitivity to NO_2_, S [%]	Response Time t_res_, [min]	Recovery Time t_rec_, [min]
LaCoO_3_	230	1.0	4.5	29	30
	300	1.1	11.9	28	30
	350	1.1	14.1	29	30
	440	1.0	5.2	29	31
La_0.9_Sr_0.1_CoO_3_	230	1.1	9.7	30	29
	300	1.1	5.5	30	29
	350	1.0	0.6	29	29
La_0.8_Sr_0.2_CoO_3_	230	1.3	30.5	30	29
	300	1.2	24.8	30	29
	350	1.2	23.7	30	29
	440	1.1	7.5	30	29

**Table 7 materials-18-01175-t007:** La(Sr)FeO_3_ thin film NO_2_-sensing response (Resp), sensitivity (S), response (t_res_), and recovery (t_rec_) times measured for 230 ÷ 440 °C.

Thin Film	Temperature T [°C]	NO_2_ Sensing Response, Resp	Thin Film Sensitivity to NO_2_, S [%]	Response Time t_res_, [min]	Recovery Time t_rec_, [min]
LaFeO_3_	230	0.3	67.0	21	9
	300	0.4	61.0	25	11
	350	1.0	3.8	5	5
	400	2.7	75.0	19	14
	440	1.7	68.1	23	12
La_0.9_Sr_0.1_FeO_3_	230	1.0	4.8	14	15
	300	1.1	8.6	15	14
	350	1.1	7.2	15	15
La_0.8_Sr_0.2_FeO_3_	230	1.2	17.9	15	15
	300	1.1	14.4	15	15
	350	1.1	9.7	16	15

## Data Availability

Data are contained within the article.

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
