# Peer review of "High-Quality Perovskite Thin Films for NO2 Detection: Optimizing Pulsed Laser Deposition of Pure and Sr-Doped LaMO3 (M = Co, Fe)"

_materials, 2025, doi:10.3390/ma18051175_

Round 1

Reviewer 1 Report

Comments and Suggestions for Authors

The paper has a good scientific soundness, and deserves to be published after major revisions, as follows:

Lack of discussion on long-term stability of thin films in real-world conditions (Lines 24-26, 621-624): The paper does not provide data on how the thin films degrade over extended periods under varying environmental conditions. A solution would be to include experimental results or a discussion on their long-term stability.

Limited exploration of competing materials for NO₂ sensing (Lines 10-26, 594-595): The study does not compare the performance of Sr-doped LaCoO₃ and LaFeO₃ thin films with other state-of-the-art materials. A comparative analysis should be included to position these materials against existing alternatives.

Absence of a clear explanation of response and recovery time variations (Lines 528-530, 540-545): The reasons behind the variation in response and recovery times, especially for Sr-doped samples, are not well explained. Providing a mechanistic discussion, possibly linking to adsorption-desorption kinetics, would strengthen the findings.

Unclear justification for the choice of doping levels (Sr 10% and 20%) (Lines 124-126, 537-538): The rationale for selecting these specific Sr doping levels is not provided. The authors should reference prior studies or include preliminary data supporting their choice.

Limited discussion on the practical implementation of the sensors (Lines 594-610): The study does not elaborate on the integration of the thin films into sensor devices, fabrication scalability, or cost-effectiveness. Addressing these aspects would enhance the paper’s applicability.

No discussion on cross-sensitivity to other gases or environmental factors (Lines 492-525): The study does not explore how these materials respond to gases like CO, NH₃, or SO₂, nor does it address the influence of humidity or temperature fluctuations. Including or referencing cross-sensitivity studies would improve the practical relevance of the findings.

Author Response

Author's Reply to the Review Report (Reviewer 1)

Thank you for your review of the submitted manuscript and your suggestions regarding the research. I trust that the explanations presented here will clear up any doubts.

Comments 1: Lack of discussion on long-term stability of thin films in real-world conditions (Lines 24-26, 621-624): The paper does not provide data on how the thin films degrade over extended periods under varying environmental conditions. A solution would be to include experimental results or a discussion on their long-term stability.

Response 1: In response to this comment, we have added a chapter (5. Summary and Discussion). Thank you for bringing this to our attention.

(…) A key issue, especially in the context of the long-term use of strontium (Sr) doped LaCoO₃ and LaFeO₃ thin films as gas sensors, is their degradation and changes in catalytic properties occurring during exposure. The degradation process is complex and depends on many factors, some of which can be relatively easily eliminated, while others require more rigorous solutions. The main factors include environmental conditions such as temperature, atmosphere, and thermal cycling. High temperatures, often exceeding the recommended operating range of the material, accelerate degradation processes including ion diffusion, phase changes, and oxidation. Moisture, COâ‚‚, SOâ‚“, and other corrosive gases can react with perovskites, leading to adverse and unpredictable changes in their composition and structure. Repeated temperature changes can generate mechanical stresses, initiating cracking and delamination of the layers.

Introducing Sr doping into LaCoO₃ and LaFeO₃ affects structural stability and allows control of oxygen ion mobility by optimizing dopant concentration, which translates to the generation of oxygen vacancies responsible for chemical activity. Thinner layers are unfortunately more susceptible to degradation due to the larger contact surface with the environment. Therefore, ensuring high-quality internal structure, i.e., a dense, oriented columnar crystal structure free of defects and discontinuities, achieved through controlled epitaxial growth, is crucial. Such conditions can be achieved using the detailed pulsed laser deposition (PLD) process, as confirmed by material research results (XRD, XPS, SEM, TEM, AFM).

The degradation rate of gas-sensitive thin films also depends on the nature of catalytic reactions. In some cases, contaminants may deposit on the perovskite surface, blocking active sites and leading to catalyst deactivation. Typical degradation processes include ion diffusion, reactions with gases, phase changes, sintering, and poisoning. At high temperatures, oxygen and metal ions can diffuse within the layer or to its surface, leading to changes in chemical composition and defect formation. Perovskites, despite their thermal stability, can react with moisture, COâ‚‚, or other gases, forming carbonates, hydroxides, or other compounds that alter their structure and properties. Chemical substances from reaction gases can bind to active sites on the material surface, poisoning the gas-sensitive layer. Excessively high temperatures can cause grain sintering, especially with significant surface porosity, which reduces the active surface area of the catalyst.

These phenomena often manifest as a significant decrease in catalytic activity, deterioration of selectivity and stability, which shortens the layer's lifespan. To effectively prevent or limit the degradation of La(Sr)CoO₃ and La(Sr)FeO₃ gas-sensitive thin films, it is necessary to optimize the chemical composition and structure of the perovskites, control environmental conditions during catalytic reactions (which is difficult to achieve), and use appropriate dopants that improve structural stability.

Considering the above, the material analyses and structural observations conducted in this work, leading to the determination of optimal PLD process parameters for producing high-quality La(Sr)CoO₃ and La(Sr)FeO₃ thin films, are extremely valuable (…)

Comments 2: Limited exploration of competing materials for NO₂ sensing (Lines 10-26, 594-595): The study does not compare the performance of Sr-doped LaCoO₃ and LaFeO₃ thin films with other state-of-the-art materials. A comparative analysis should be included to position these materials against existing alternatives.

Response 2: In response to this comment, we have added Chapter 5, titled "Summary and Discussion". A detailed explanation is provided below. Thank you for bringing this to our attention.

(…) In addition to the perovskites discussed in this paper, several other thin-film materials are being intensively studied for use in gas sensors, including NOâ‚‚. These primarily include simple metal oxides (SnOâ‚‚, TiOâ‚‚, Inâ‚‚O₃, WO₃), known for their high sensitivity to various gases, including NOâ‚‚. They can be fabricated into thin films with controlled structures using various techniques, allowing for the adjustment of their catalytic properties. Graphene, molybdenum disulfide, and other transition metal halides are also promising. These materials, due to their large specific surface area and unique electronic properties, show potential in gas detection.

Perovskites, including La(Sr)CoO₃ and La(Sr)FeO₃, which are the subject of this research, often exhibit improved sensitivity to a broad range of gases, including volatile organic compounds (VOCs), nitrogen oxides, and carbon monoxide. Their crystal structure and surface properties facilitate strong interactions with gas molecules, leading to significant changes in electrical conductivity. In contrast, traditional metal oxides may have limited sensitivity to certain gases or require higher operating temperatures, which often restricts their application in sensors. Furthermore, by appropriately adjusting the perovskite composition (precise doping), its selectivity towards specific gases can be tuned. This tunability stems from the ability to alter the oxygen binding energy and create oxygen vacancies in the crystal lattice, which selectively interact with specific gas molecules. Achieving high selectivity with traditional metal oxides can be challenging due to their lower compositional and structural flexibility.

Perovskite-based sensors can operate at lower temperatures compared to traditional oxide sensors. This advantage reduces energy consumption and improves sensor lifespan. Traditional metal oxides often require high temperatures to achieve optimal performance, leading to energy consumption and stability issues. Perovskites often exhibit faster response times to changes in gas concentration. This rapid response is crucial in applications requiring real-time monitoring. Traditional oxide sensors may have slower response times, limiting their usefulness in high dynamic range applications. The perovskite structure allows for a vast variety of compositions by incorporating different metals and non-metals. This diversity enables the tailoring of sensor properties to meet the requirements of specific applications. Traditional metal oxides tend to have more fixed compositions, offering less flexibility in sensor design (…)

Comments 3: Absence of a clear explanation of response and recovery time variations (Lines 528-530, 540-545): The reasons behind the variation in response and recovery times, especially for Sr-doped samples, are not well explained. Providing a mechanistic discussion, possibly linking to adsorption-desorption kinetics, would strengthen the findings.

Response 3: Thank you for pointing this out. The following discussion has been added to Chapter 4.6. “Electrical Resistance Measurements” to clarify this issue

(…) Variations in the reaction and regeneration times of Sr-doped LaCoO₃ and LaFeO₃ perovskite thin films arise from a complex interplay between the material's structure (surface), adsorption-desorption mechanisms, and the reaction kinetics with the active gas. Strontium doping significantly alters these properties by generating oxygen vacancies (with concentrations closely correlated to strontium content) and modifying the cation oxidation states. Given a structural analysis of the synthesized perovskite thin films, and assuming a consistent interaction mechanism of surface oxygen defects at the active sites across all samples, the disparities in response and regeneration times can be attributed to surface topography. The observed structural fragmentation in La(Sr)FeO₃ compared to La(Sr)CoO₃ (as evidenced by SEM and AFM images, and measured roughness parameters) appears to correspond to a reduction in response and regeneration times. Decreasing crystallite size increases the concentration of active centers, thereby substantially stabilizing the sensitivity of thin films. As demonstrated, optimizing the structure of the gas-sensitive material is essential for maintaining high catalytic performance (…)

Comments 4: Unclear justification for the choice of doping levels (Sr 10% and 20%) (Lines 124-126, 537-538): The rationale for selecting these specific Sr doping levels is not provided. The authors should reference prior studies or include preliminary data supporting their choice.

Response 4: There is no strict minimum strontium (Sr) content required to induce a doping effect in LaCoO₃ and LaFeO₃ perovskites. This effect depends on a number of factors, including: the type of material (LaCoO₃ and LaFeO₃ exhibit different properties, which affects their response to strontium doping), temperature (doping effects may be more pronounced at higher temperatures), synthesis method (the way the material is fabricated can affect the distribution of dopants and their effectiveness), and crystal structure (defects and inhomogeneities in the crystal structure, as well as differences in the ionic radii of Sr, Co, and Fe, can affect the observed doping effects). However even small amounts of Sr (on the order of a few atomic percent) can already cause significant changes in the electrical and magnetic properties of these perovskites. On the other hand, excessively high Sr doping concentrations should be avoided, as the generation of excessive cationic vacancies associated with this can significantly reduce conductive properties. Studies show that doping effects are more pronounced in LaCoO₃ than in LaFeO₃. This is due to differences in the ionic/electronic structure and initial chemical properties of these materials.

Relevant clarifications to the issue raised are included in the revised text:

(…) Modeling the electrical resistance of La(Sr)CoO₃ and La(Sr)FeO₃ perovskite thin films with varying strontium (Sr) doping concentrations presents a complex challenge. A criti-cal consideration involves changes in the ionic structure, particularly the creation of oxy-gen vacancies. These alterations can compromise the integrity of the base perovskite, even at low Sr doping levels (a few atomic percentages). In this study, we deliberately selected 10% and 20% Sr doping to produce targets with precise stoichiometric compositions: La0.9Sr0.1CoO3, La0.8Sr0.2CoO3, La0.9Sr0.1FeO3, La0.8Sr0.2FeO3 as proven by the material analysis presented above (XRD, SEM/EDS, TEM, AFM and XPS).

Initially, obtaining significant and reproducible resistance (conductivity) changes at low dopant concentrations (1 ÷ 5% Sr) proves difficult, as substrate influence can obscure measurements. Conversely, excessive Sr doping (approaching 40 ÷ 50% Sr and above) induces substantial distortions in the cationic sublattice and increases the risk of surface defects [48]. These defects can generate unfavorable tensile stresses, outweighing the benefits of increased surface chemically active centers (relevant to catalysis). Consequently, measurements become dominated by structural changes, introducing significant errors. The chosen Sr doping concentrations define a range for substantial conductivity (resistivity in the presence of NOâ‚‚) variations, enabling a planned, progressive narrowing of these ranges in subsequent studies.

Increased strontium concentration influences charge carrier hopping between cobalt (Co) or iron (Fe) ions of varying valences, thus modifying resistance (as discussed in XRD studies). At higher temperatures (above 500°C), polaron transport may become dominant, especially in highly defective materials (with defect concentrations up to several tens of percent). These observations suggest an optimal strontium doping range of a few to ap-proximately 15%, which balances high conductivity with structural compactness, homogeneity, and controlled surface roughness. Exceeding or falling below this range can lead to uncontrolled increases in resistance (…)

Comments 5: Limited discussion on the practical implementation of the sensors (Lines 594-610): The study does not elaborate on the integration of the thin films into sensor devices, fabrication scalability, or cost-effectiveness. Addressing these aspects would enhance the paper’s applicability.

Response 5: Thank you for pointing this out. The following discussion has been added to Chapter 5. “Summary and discussion” to clarify this issue.

(…) Perovskite sensors utilizing pure and Sr-doped LaCoO3 and LaFeO3 thin films demonstrate significant potential across diverse applications, particularly in gas sensing (detecting NOx, CO, VOCs, and other gases) and environmental monitoring (air quality assessment, industrial safety systems, and medical diagnostics). Strontium (Sr) doping enhances the conductivity and reactivity of these materials, leading to improved sensitivity and selectivity. Furthermore, the temperature-dependent electrical resistance of these films enables their application in temperature sensors, while their sensitivity to humidity variations opens avenues for humidity sensors used in environmental monitoring.

These LaCoO3 and LaFeO3 perovskite thin films can be fabricated through various methods, including magnetron sputtering, atomic layer deposition (ALD), molecular beam epitaxy (MBE), and pulsed laser deposition (PLD) or pulsed electron deposition (PED), both of which are PVD-based ablation techniques. Ablation techniques, unlike others, minimize gas-phase decomposition and, when optimized, ensure precise stoichiometry and chemical/phase composition transfer from target to substrate. Ablation methods, offering precise thickness control through adjustments in parameters such as power density, working distance, and pulse count, are particularly well-suited for producing La(Sr)CoO₃ and La(Sr)FeO₃ thin films. This precision facilitates seamless integration into microelectromechanical systems (MEMS), enabling miniaturization and integration with other electronic components.

These fabrication techniques are scalable and relatively cost-effective, promoting the mass production of thin films. Ongoing advancements in ablation, ALD, and MBE technologies allow for highly accurate control of layer thickness and composition, crucial for applications demanding high precision and dimensional repeatability. The low cost of starting perovskite materials further enhances the economic viability of mass production. Moreover, the development of fabrication techniques on flexible substrates paves the way for wearable sensor devices, integrating microelectronic sensors into everyday apparel.

Comments 6: No discussion on cross-sensitivity to other gases or environmental factors (Lines 492-525): The study does not explore how these materials respond to gases like CO, NH₃, or SO₂, nor does it address the influence of humidity or temperature fluctuations. Including or referencing cross-sensitivity studies would improve the practical relevance of the findings.

Response 6: The research presented in this publication intentionally focused on a single active agent – NO2. Concurrently (in part), studies were conducted to determine the effect of strontium doping on the catalytic properties of lanthanide thin films from the LaCoO3 and LaFeO3 families. Based on this, among other factors, it was decided to place greater emphasis on the structure of thin films, including both surface topography and internal structure. The research results presented in this summary primarily reflect efforts to obtain the highest quality thin films, which is neither easy nor reproducible in ablation processes. As indicated by the publication's title, the structure, or more precisely, a thorough and comprehensive material analysis, is crucial in this case. Measurements of catalytic properties are time-consuming, sensitive to a range of external factors, and require precisely prepared and tested samples. Therefore, the main task of the research team became formulating optimized guidelines, considering the parameters of the laser ablation process, which will be used in the next step for the reproducible deposition of high-quality La(Sr)CoO3 and La(Sr)FeO3 thin films. Currently, the obtained and analyzed set of parameters can be successfully used to produce high-quality functional thin-film materials, both ceramic and composite (including nanometric).

Cross-correlation studies of the catalytic sensitivity of La(Sr)CoO3 and La(Sr)FeO3 thin films in the presence of other (mentioned by the reviewer) deleterious agents are currently underway and will certainly be published. Compiling such a large amount of data (results) in a single paper, in the authors' opinion, would be too extensive and could hinder the correct assessment in formulating the proper relationship between the deliberate changes in internal structure (ionic and electronic) due to Sr doping and the significant (often subtle) changes in the catalytic activity of La(Sr)CoO3 and La(Sr)FeO3 thin films.

Reviewer 2 Report

Comments and Suggestions for Authors

This article describes the high-quality perovskite thin films for NO₂ detection using the optimizing pulsed laser deposition of pure and Sr-doped LaMO₃ (M = Co, Fe).

To improve the manuscript, the authors should consider the following modifications:

(1) The authors presented the analysis of the structure and properties of pure and Sr-doped LaCoO3 and LaFeO3 thin films for use in resistive sensors specifically for NO2 detection. The authors should discuss the detection of harmful gases such as Nitric oxide (NO), Nitrous oxide (N2O), Carbon monoxide (CO), Hydrogen (H2), Ammonia (NH3), Sulfur dioxide (SO2), and Hydrogen sulfide  (H2S) using the pure and Sr-doped LaCoO3 and LaFeO3 thin films.

(2) The authors presented the selected mechanical properties and adhesion results obtained for La(Sr)CoO3 and  La(Sr)FeO3 thin films in Table 5. This manuscript lacks an error analysis of the apparent viscosity value which is highly needed for readability purposes. The authors should place the standard deviations of the Nano-hardness HV; Indentation hardness HIT, (GPa); Young’s Modulus EIT, (GPa); Penetration depth hm, (nm); and Critical load Lc, (mN) of the La(Sr)CoO3 and  La(Sr)FeO3 thin films in Table 5 so that the reader will have an idea of the reproducibility of the data.

(3) The authors presented the “4.6. Electrical Resistance Measurements” section, however, the authors should discuss the modeling aspects of the electrical resistance of the La(Sr)CoO3 and  La(Sr)FeO3 thin films with varied dopant concentrations.

The submitted manuscript contains significant scientific insights, and the experimental and molecular conformation data support the conclusions. However, the manuscript requires minor revisions before being accepted in the well-circulated journal, Materials.

Author Response

Comments 1: The authors presented the analysis of the structure and properties of pure and Sr-doped LaCoO3 and LaFeO3 thin films for use in resistive sensors specifically for NO2 detection. The authors should discuss the detection of harmful gases such as Nitric oxide (NO), Nitrous oxide (N2O), Carbon monoxide (CO), Hydrogen (H2), Ammonia (NH3), Sulfur dioxide (SO2), and Hydrogen sulfide (H2S) using the pure and Sr-doped LaCoO3 and LaFeO3 thin films.

Response 1: The research presented in this publication intentionally focused on a single active agent – NO2. Concurrently (in part), studies were conducted to determine the effect of strontium doping on the catalytic properties of lanthanide thin films from the LaCoO3 and LaFeO3 families. Based on this, among other factors, it was decided to place greater emphasis on the structure of the thin films, including both surface topography and internal structure. The research results presented in this summary primarily reflect efforts to obtain the highest quality thin films, which is neither easy nor reproducible in ablation processes. As indicated by the publication's title, the structure, or more precisely, a thorough and comprehensive material analysis, is crucial in this case. Measurements of catalytic properties are time-consuming, sensitive to a range of external factors, and require precisely prepared and tested samples. Therefore, the main task of the research team became formulating optimized guidelines, taking into account the parameters of the laser ablation process, which will be used in the next step for the reproducible deposition of high-quality La(Sr)CoO3 and La(Sr)FeO3 thin films. Currently, the obtained and analyzed set of parameters can be successfully used to produce high-quality functional thin-film materials, both ceramic and composite (including nanometric).

Cross-correlation studies of the catalytic sensitivity of La(Sr)CoO3 and La(Sr)FeO3 thin films in the presence of other (mentioned by the reviewer) deleterious agents are currently underway and will certainly be published. Compiling such a large amount of data (results) in a single paper, in the authors' opinion, would be too extensive and could hinder the correct assessment in formulating the proper relationship between the deliberate changes in internal structure (ionic and electronic) due to Sr doping and the significant (often subtle) changes in the catalytic activity of La(Sr)CoO3 and La(Sr)FeO3 thin films.

Comments 2: The authors presented the selected mechanical properties and adhesion results obtained for La(Sr)CoO3 and La(Sr)FeO3 thin films in Table 5. This manuscript lacks an error analysis of the apparent viscosity value which is highly needed for readability purposes. The authors should place the standard deviations of the Nano-hardness HV; Indentation hardness HIT, (GPa); Young’s Modulus EIT, (GPa); Penetration depth hm, (nm); and Critical load Lc, (mN) of the La(Sr)CoO3 and La(Sr)FeO3 thin films in Table 5 so that the reader will have an idea of the reproducibility of the data.

Response 2: Apparent viscosity analysis is particularly significant in the context of liquid nanomaterials and their associated technological processes for nanolayer formation. For solid nanolayers, its importance is more indirect, relating to the impact of application processes on structure and mechanical properties. This study focuses exclusively on ceramic materials (doped perovskites), which are inherently hard and brittle, and therefore not susceptible to deformation. The irregularities observed in the curves representing nanomechanical property measurements (NHT, NST – Figure 11) are a direct result of fracture in the tested thin films upon contact with the Berkovich indenter. They cannot, in any way, be attributed to the concept of apparent viscosity. Consequently, focusing on the deposition process itself via the PLD technique, using identical substrates (Si, MgO) with known orientation and subjected to thorough surface cleaning, apparent viscosity was not analyzed. In the chapter dedicated to the analysis of thin film cross-sectional structure using TEM imaging (Figures 12 ÷ 15), it is clearly evident that there are no effects at the fim-substrate interface suggesting any abnormal interactions resulting from deformation or plastic flow.

Based on the research team's previous experience with the application of ablation techniques (PLD, PED) to thin films of materials such as Co,Ni(Ca)O, Bi2O3, LaCeO3, LaCaCoO3, YSZ, and a range of composite coatings, no significant effect of apparent viscosity on either the structure or, most importantly, their nanomechanical (tribological) properties was observed. With such thin materials, we are essentially dealing with a substrate-layer system, where the substrate itself significantly determines the aforementioned properties. Nanomechanical property measurements in the presented studies were conducted with careful adherence to the guidelines of relevant standards (penetration depth, dwell times, load range, scratch length, indentation spacing, etc.). Each result presented in Table 5 is an average value from a complete series of measurements (at least 15 per series), accounting for the instrument's measurement errors (CSM Instruments). As rightly suggested by the reviewer, the relevant standard deviation values for the mentioned data have been included.

As suggested by the reviewer, the standard deviation values for individual (averaged) measurements of nanomechanical properties and adhesion have been supplemented in Table 5 with the above data.

Comments 3: The authors presented the “4.6. Electrical Resistance Measurements” section, however, the authors should discuss the modeling aspects of the electrical resistance of the La(Sr)CoO3 and La(Sr)FeO3 thin films with varied dopant concentrations.

Response 2: Modeling the electrical resistance of La(Sr)CoO₃ and La(Sr)FeO₃ perovskite thin films, with varying strontium (Sr) doping concentrations, presents a multifaceted challenge. The primary consideration involves changes in the ionic structure, specifically the generation of oxygen vacancies. These alterations can compromise the base perovskite's integrity, even at low Sr doping levels (a few atomic percentages). In this study, we deliberately selected 10% and 20% Sr doping to produce targets with precise stoichiometric compositions.

Initially, obtaining significant and reproducible resistance (conductivity) changes at low dopant concentrations (1 - 5% Sr) proves difficult, with substrate influence potentially obscuring measurements. Conversely, excessive Sr doping (approaching 40 - 50% Sr or above) induces substantial distortions in the cationic sublattice and increases the risk of surface defects [44]. These defects can generate unfavorable tensile stresses, outweighing the benefits of increased surface chemically active centers (relevant to catalysis). Consequently, measurements become dominated by structural changes, introducing significant error. The chosen Sr doping concentrations define a range for substantial conductivity (resistivity in the presence of NOâ‚‚) variations, facilitating a planned, progressive narrowing of these ranges in subsequent studies.

In thin films, grain boundaries impede electron flow, elevating resistance. To mitigate this, we conducted detailed structural analyses to develop a deposition procedure yielding defect-free films with structures corresponding to zones I and T of Thornton's model. Surface topography and cross-sectional internal structure evaluations revealed the impact of Sr doping on structural modifications.

Hole conduction, associated with valence band holes, dominates conductivity in La(Sr)CoO₃ and La(Sr)FeO₃. Increased Sr concentration influences charge carrier hopping between cobalt (Co) or iron (Fe) ions of varying valences, modifying resistance (as discussed in the XRD studies). At higher temperatures (above 500°C), polaron transport may become dominant, particularly in highly defective materials (up to several tens of percent defects). These observations suggest an optimal Sr doping range of a few to approximately 15%, balancing high conductivity with structural compactness, homogeneity, and controlled surface roughness. Exceeding or falling below this range can lead to uncontrolled resistance increases.

External factors, notably temperature, significantly influence La(Sr)CoO₃ and La(Sr)FeO₃  electrical resistance. Our modeling incorporates this dependence, hence the 230 ÷ 440°C measurement range. While limited, this range aligns with the observed reproducible changes in catalytic properties upon exposure to specific external agents. This consideration is crucial for potential applications in resistive sensors, where size and weight constraints necessitate lightweight, chemically resistant support materials (e.g., light non-ferrous metal alloys or polymer composites). This practical requirement restricts the operational temperature to below 500°C, justifying our chosen range.

Common methods for modeling thin-film electrical resistance include density functional theory (DFT) for electronic structure and defect calculations, Monte Carlo simulations for charge carrier transport, equivalent circuit models for grain boundary and defect analysis, and microscopic techniques (SEM, TEM, AFM) employed in this study. This research focuses on verifying real-world film growth conditions, aiming for idealized structures that correlate with predicted catalytic property changes in the presence of NOâ‚‚ within the specified temperature range, rather than predictive modeling of structural and property variations.

To optimize the electrical properties of La(Sr)CoO₃ and La(Sr)FeO₃ thin films for electronic and electrochemical device applications, a strong understanding of these factors is required.

Based on the above observations and suggestions, a section has been added to chapter “4.6. Electrical Resistance Measurements”

Reviewer 3 Report

Comments and Suggestions for Authors

High-Quality Perovskite Thin Films for NOâ‚‚ Detection:  Optimizing Pulsed Laser Deposition of pure and Sr-Doped LaMO₃ (M = Co, Fe) is very interesting paper written on 31 pages. Minor improvements are required. The modern high effective characterization methods offer many information about High-Quality Perovskite Thin Films characteristics.

Line 16: nanostructured surfaces with varying morphologies (such as….)

Line 21, 22: Sr doping was found to stabilize the catalytic activity of LaFeO3, although its behavior in the presence of the oxidizing agent NO2 differed from that of LaCoO3 (in what temperature range?)

Line 54, 55: The  energy industry also needs gas sensors, especially in power plants where high temperatures and pressures require durable and reliable sensors (such as…). In range of pressure are these sensors mostly used?

Line 113, 114: At low temperatures, the oxide exhibits the characteristics of a non-magnetic insulator, while at high temperatures it exhibits the characteristics of a metallic paramagnet (what are high temperatures in this case?)

Line 171, 172: This requires considerable experience in selecting the appropriate substrate and, more importantly, optimizing several key PLD process parameters (such as…)

Line 187 The resulting nanopowders (particle size and morphology??)

Line 212: and simultaneously increasing the evaporation time (what is maximal evaporation time?)

Line 295, 296: The stronger doublet at a lower binding energy (BE ≈ 780 eV) is  the direct photoelectron line, while the satellite doublet is formed through a shake-up process (please to write most important parameters for shake-up process)

Line 340: Operating at excessively low energy densities necessitates more pulses  to achieve the desired coating thickness (what is maximal coating thickness?)

Line 545: were observed across the analyzed temperature range (such as..)

Conclusion:

Line 592, 593: This study investigated the structural, morphological, and electrical properties of  pure and Sr-doped LaCoO3 and LaFeO3 thin films deposited by pulsed laser deposition  (PLD) for potential application as sensing electrodes (SE) in NO2 gas sensors (in what temperature interval?)

Line 600: what is minimal content of Sr required for Sr Doping Effects?

General question:

  1. What is main influence of Sr in structure perovskite for NOâ‚‚ Detection at higher temperatures?
  2. Sr doping stabilized the catalytic activity of, resulting in faster response and recovery times and a more consistent response to NO2. However, increased noise at higher temperatures limited the sensitivity measurements. How to solve this problem?

Author Response

Author's Reply to the Review Report (Reviewer 3)

Thank you for your review of the submitted manuscript and your suggestions regarding the research. I trust that the explanations presented here will clear up any doubts.

Comments 1: Line 16: nanostructured surfaces with varying morphologies (such as….)

Response 1: The author's reference was to the topographical changes highlighted in subsequent material analysis. This analysis examined surface development, size, and the specific shape of surface irregularities. In comparing LaCoO3 and LaFeO3, the analysis considered internal structures - columnar, compact crystal. This comparison suggests potential differences in the morphology of elements observed on the films' surfaces.

Relevant information has been added to the abstract.

Comments 2: Line 21, 22: Sr doping was found to stabilize the catalytic activity of LaFeO3, although its behavior in the presence of the oxidizing agent NO2 differed from that of LaCoO3 (in what temperature range?)

Response 2: This statement refers directly to results obtained only for the pure LaFeO3 thin film and its Sr-doped variants. The graphs and data table show that, within the narrow temperature range of 300 ÷ 350°C, response and recovery times remain consistent, and the perovskites exhibit clear sensitivity and response. Further stabilization of these values might be observed if measurements at higher temperatures (400°C and above) were not affected by emerging noise. We are planning to carry out such measurements soon and to verify the data we have obtained.

Relevant information has been added to the abstract.

Comments 3: Line 54, 55: The  energy industry also needs gas sensors, especially in power plants where high temperatures and pressures require durable and reliable sensors (such as…). In range of pressure are these sensors mostly used?

Response 3: We are discussing industrial sensors based on proven materials. Industrial sensors are components of automation systems, tasked with detecting and recording environmental signals crucial for control algorithms. Currently, they are indispensable for the operation of most machinery and production lines. They can be viewed as the sensory organs of technology. Regardless of the complexity of the requirements, properly selected sensors can meet numerous challenges, becoming a reliable solution in many applications. Among the most commonly used are proximity sensors (including capacitive, inductive, and photoelectric sensors). Each employs a distinct operating principle, and their selection and application depend on the specific application and operating environment.

There is no single, universal operating pressure range. It is highly dependent on the specific application within a power plant. However, it's safe to say that power plants utilize gas sensors capable of withstanding a very broad pressure range (for example: natural gas pipeline pressures can vary significantly, from a few hundred Pa to over 5000 Pa. For thin film applications, gas concentrations can range from several to several hundred ppm.

The relevant link is attached to the text.

Comments 4: Line 113, 114: At low temperatures, the oxide exhibits the characteristics of a non-magnetic insulator, while at high temperatures it exhibits the characteristics of a metallic paramagnet (what are high temperatures in this case?)

Response 4: As the temperature rises, a portion of the cobalt atoms transitions to a high-spin state, where their magnetic moments become unpaired. This transformation renders the material paramagnetic, meaning it responds to an external magnetic field but does not retain permanent magnetization once the field is removed. Furthermore, the elevated temperature provides sufficient energy for electrons to jump to the conduction band, imparting metallic properties to the material. For LaCoO₃, "high temperature" refers to a temperature range exceeding approximately 500 K (about 227°C). The precise temperature of the phase transition can vary slightly, depending on synthesis conditions and material purity.

Comments 5: Line 171, 172: This requires considerable experience in selecting the appropriate substrate and, more importantly, optimizing several key PLD process parameters (such as…)

Response 5: The core challenge lies in the intricate interplay of several parameters: the energy of individual laser pulses, which dictates the power density at the target surface (factoring in optical path losses), the target-substrate distance, the working gas pressure within the chamber, and the substrate temperature. Changing any of these can significantly affect the others, which then changes how ablation, plasma material transport, and substrate crystal growth work. To get high-quality crystal films growing correctly on the substrate, you need to look at all these factors together.

Relevant information has been added to the text.

Comments 6: Line 187 The resulting nanopowders (particle size and morphology??)

Response 6: In the mechanical synthesis process employed here, utilizing a planetary mill (reactive grinding) at a precisely selected energy, determined by a rotational speed of approximately 650 RPM, powder can be reduced to nanometer-scale dimensions. As this publication focuses on the synthesis of base oxides rather than the grinding process itself, a detailed characterization of the final material is intentionally omitted. An example SEM image of La(Sr)CoO3 powder, obtained after the complete process, is presented below, clearly illustrating the morphology of the resulting crystallite clusters and allowing for verification of their size.

Figure 1. SEM image of La(Sr)CoO3 powder obtained after complete mechanical synthesis process

Comments 7: Line 212: and simultaneously increasing the evaporation time (what is maximal evaporation time?)

Response 7: With laser ablation, the pulse duration stays the same for a specific wavelength (harmonic), no matter the energy. In this study, using the fourth harmonic (266 nm), the laser had a pulse duration of about 4 ÷ 6 ns.

Estimating the maximum vaporization time is not typically performed, as it is directly derived from the number of pulses and dictates the thickness of the resulting thin film for a given pulse energy. It is important to note that Pulsed Laser Deposition (PLD) is primarily a technique for producing thin films, generally defined as those with thicknesses up to 200 nm. While ablation can be used to create thicker films (e.g., up to 500 nm), this often leads to a decline in film quality, characterized by disrupted or unstable growth, and the formation of deltoid structures instead of columnar crystals.

Comments 8: Line 295, 296: The stronger doublet at a lower binding energy (BE ≈ 780 eV) is  the direct photoelectron line, while the satellite doublet is formed through a shake-up process (please to write most important parameters for shake-up process)

Response 8: The shake-up process involves the excitation of valence electrons to a higher energy level. This occurs at the expense of some of the energy of the X-ray photons used to excite photoelectrons from core levels. This phenomenon is particularly common in transition metal oxides. Consequently, some photoelectrons ejected from the Co 2p core level possess lower kinetic energy, which corresponds to higher binding energy. This results in the creation of a satellite line, shifted relative to the main line by a specific value (approximately 5.5 eV in this case, representing the valence electron excitation energy), and with an intensity that is a fraction of the main line (approximately 0.2, depending on the proportion of photons causing the shake-up effect). The specifics of the shake-up process are generally disregarded when the primary objective is the chemical determination of chemical states. The presence of a satellite peak merely confirms the oxidation state of the metallic element.

Comments 9: Line 340: Operating at excessively low energy densities necessitates more pulses to achieve the desired coating thickness (what is maximal coating thickness?)

Response 9: The determination of film thickness achieved through the PLD technique has been discussed previously (Response 7). Thin films, in this context, are defined as those with thicknesses generally not exceeding approximately 200 nm.

Comments 10: Line 545: were observed across the analyzed temperature range (such as..)

Response 10: This refers to the temperature range of 230 ÷ 440°C, as presented in the figures and data tables.

The relevant information is detailed within the publication text.

Comments 10: Line 592, 593: This study investigated the structural, morphological, and electrical properties of pure and Sr-doped LaCoO3 and LaFeO3 thin films deposited by pulsed laser deposition (PLD) for potential application as sensing electrodes (SE) in NO2 gas sensors (in what temperature interval?)

Response 10: This study investigates the temperature range of 230°C to 440°C. Future research will expand this range to include temperatures starting from ambient levels.

Relevant information has been added to the text.

Comments 11: Line 600: what is minimal content of Sr required for Sr Doping Effects?

Response 11: There is no strict minimum strontium (Sr) content required to induce a doping effect in LaCoO₃ and LaFeO₃ perovskites. This effect depends on a number of factors, including: the type of material (LaCoO₃ and LaFeO₃ exhibit different properties, which affects their response to strontium doping), temperature (doping effects may be more pronounced at higher temperatures), synthesis method (the way the material is fabricated can affect the distribution of dopants and their effectiveness), and crystal structure (defects and inhomogeneities in the crystal structure, as well as differences in the ionic radii of Sr, Co, and Fe, can affect the observed doping effects).

However even small amounts of Sr (on the order of a few atomic percent) can already cause significant changes in the electrical and magnetic properties of these perovskites. On the other hand, excessively high Sr doping concentrations should be avoided, as the generation of excessive cationic vacancies associated with this can significantly reduce conductive properties.

Studies show that doping effects are more pronounced in LaCoO₃ than in LaFeO₃. This is due to differences in the ionic/electronic structure and initial chemical properties of these materials.

Comments 12: General question:

  1. What is main influence of Sr in structure perovskite for NOâ‚‚ Detection at higher temperatures?

Response 12: According to the authors, the primary factor determining the impact of strontium (Sr) doping on the catalytic properties of the analyzed lanthanides at elevated temperatures is the modification of the crystal structure, and consequently, the ionic/electronic structure (specifically, the presence of cationic vacancies). As stated in the text, introducing Sr dopants alters the perovskite's crystal structure, inducing defects (vacancies). These alterations increase the number of active sites on the surface, thereby enhancing the measured catalytic activity. Sr doping generates oxygen vacancies, which are particularly active in the oxidation of organic compounds.

Simultaneously, the dopants can modify the conduction and valence bands of the perovskite, affecting its ability to absorb light and generate charge carriers. This is crucial for photocatalytic applications. Furthermore, the modification of the Fermi level by dopants can influence electron transfer at the surface, a key factor in redox reactions.

In certain scenarios, dopants can segregate on the perovskite surface (for example, if they are pushed to the forefront during gas-phase crystallization), altering its chemical composition and surface properties. This can lead to the formation of new active sites or the modification of existing ones. Finally, these dopants can also affect the perovskite's ability to adsorb molecules from the surrounding environment. These changes in surface structure and electronic properties can influence the strength and type of interaction between the perovskite surface and the adsorbed molecules.

In the present study, the effects related to segregation and concentration variations at the surface were eliminated (as confirmed by XRD, XPS, and EDS analyses in SEM and TEM chemical/phase analyses) through optimization of the PLD deposition process. Consequently, the key catalytic properties (NO2 detection capability) should be attributed solely to the modification of the crystalline and ionic structure (the presence of cationic vacancies).

Comments 13: General question:

  1. Sr doping stabilized the catalytic activity of, resulting in faster response and recovery times and a more consistent response to NO2. However, increased noise at higher temperatures limited the sensitivity measurements. How to solve this problem?

Response 13: Several solutions can be implemented to address this situation. The most practical approach appears to be modifying the microstructure, specifically the surface topography, by further reducing roughness and/or eliminating droplets and clusters on the sample surfaces. Decreasing the material's specific surface area in this manner can lead to a reduction in noise levels. Material analysis has shown that the tested thin films exhibit high quality in terms of surface area and internal structure. However, it's possible that the surface becomes contaminated during measurements or that elements of the intentionally attached measuring electrodes appear on the surface. These issues must be eliminated, and this will certainly be considered in future planned measurements of catalytic properties.

Another solution involves optimizing measurement conditions, such as employing signal filtering techniques or utilizing better shielded and temperature-stabilized measurement systems, to reduce noise.

Round 2

Reviewer 1 Report

Comments and Suggestions for Authors

Authors addressed my issues point by point.

Comments on the Quality of English Language

A good quality of English has been detected in this manuscript at the end of the second round of revisions.